# A deep learning approach for complex microstructure inference

Ali Riza Durmaz [1,2,3,7 ✉], Martin Müller[4,5,7], Bo Lei [6], Akhil Thomas [1,3], Dominik Britz [4,5], Elizabeth A. Holm[6], Chris Eberl [1,3], Frank Mücklich[4,5] & Peter Gumbsch[1,2]

Automated, reliable, and objective microstructure inference from micrographs is essential for a comprehensive understanding of process-microstructure-property relations and tailored materials development. However, such inference, with the increasing complexity of micro-structures, requires advanced segmentation methodologies. While deep learning offers new opportunities, an intuition about the required data quality/quantity and a methodological guideline for microstructure quantification is still missing. This, along with deep learning's seemingly intransparent decision-making process, hampers its breakthrough in this field. We apply a multidisciplinary deep learning approach, devoting equal attention to specimen preparation and imaging, and train distinct U-Net architectures with 30–50 micrographs of different imaging modalities and electron backscatter diffraction-informed annotations. On the challenging task of lath-bainite segmentation in complex-phase steel, we achieve accuracies of 90% rivaling expert segmentations. Further, we discuss the impact of image context, pre-training with domain-extrinsic data, and data augmentation. Network visualization techniques demonstrate plausible model decisions based on grain boundary morphology.

[1] Fraunhofer Institute for Mechanics of Materials IWM, Freiburg 79108, Germany. [2] Karlsruhe Institute of Technology (KIT), Institute for Applied Materials IAM, Karlsruhe 76131, Germany. [3] University of Freiburg, Freiburg 79110, Germany. [4] Department of Materials Science, Saarland University, Saarbrücken 66123, Germany. [5] Material Engineering Center Saarland, Saarbrücken 66123, Germany. [6] Department of Materials Science and Engineering, Carnegie Mellon University, Pittsburgh, PA 15213, USA. [7] These authors contributed equally: Ali Riza Durmaz, Martin Müller. ✉email: ali.riza.durmaz@iwm.fraunhofer.de

Deep learning (DL) is a lasting subject of attention and achieved remarkable results that culminated in a paradigm shift in computer vision. In particular, research fields such as autonomous driving[1], and biomedicine[2,3] acted as driving forces for the development of data-driven approaches, which superseded conventional computer vision (CV) algorithms to a large extent. The introduction of convolutional neural networks (CNN) with versatile architectures was accompanied by substantial improvements in CV tasks[4]. This was rendered possible by the accessibility of source codes and open-access data sets, which enabled the comparability of different modeling approaches and thus steady improvement.

Quality control in materials processing or of safety-critical components, as well as tailored materials development[5] require the segmentation and classification of material microstructures. Segmentation here refers to the pixel/voxel-wise materials phase assignment. It is indispensable when relating microstructure with properties, e.g., the phase morphology with fatigue properties[6]. Predominantly, 2D micrographs of different imaging modalities, such as light optical microscopy (LOM) or scanning electron microscopy (SEM), are utilized for microstructure inference.

However, such micrographs' automated, reliable, and objective segmentation is not established for all desirable material classes. Although DL has more than proven its capability for image segmentation and classification, it is still waiting for its breakthrough in materials science. This can be attributed to DL being frequently associated with a few drawbacks. Namely, the requirement for (very) large data quantities and the black-box nature of CNNs concerning the intransparency of their decisions[7]. Furthermore, microstructure recognition tasks, compared to natural images such as ImageNet[8], can be very complex regarding the degree of detail and information density in the images. This further impedes the determination of accurate annotations[9] needed for supervised-learning, which may discourage the use of DL, ultimately resulting in a lack of representative annotated open-access data sets.

Hence, there is no practical guide on suitable specimen preparation and contrasting, data acquisition and processing, and no general intuition about the quality and quantity of data required to train a specific DL architecture in the material science domain. Consequently, material scientists' recurrent questions address the required amount of training data, resolutions, annotation accuracy, model architectures, and training strategies.

The work's primary objective is to tackle former questions and provide a better grasp through an integral approach systematically investigating methodological interdependencies in the whole metallographic and DL process chain. Moreover, a CNN's decision-making process is rendered more transparent by investigating the importance of certain microstructural features for the CNN prediction. Using the microstructure of a complex phase (CP) steel, and particularly its lath-shaped bainitic phase, as a case study, we demonstrate DL's relevance in the field and aim to raise the awareness and acceptance of DL for such tasks. This microstructure class exhibits pronounced importance in engineering, and its constituents can only be segmented to a minimal extent using classical CV approaches[10].

According to the classification scheme suggested by Zajac[11], the microstructure of CP steels, a family of advanced high-strength steels, typically consists of bainite (granular, upper, or degenerate upper bainite), ferrite, and dispersed carbon-rich additional phases like martensite or retained austenite. In micrographs of such heterogeneous microstructures, not all constituents can be distinguished through gray value distribution. Therefore, simple, traditional segmentation methods operating on LOM or SEM quickly reach their limit. Approaches to quantify the separate microstructure constituents using EBSD individually

have been reported[12,13]. Müller et al.[14] developed a procedure to segment lath-shaped bainite in CP steel micrographs consisting of lath-shaped and granular bainite by analyzing the microstructure constituents' directionality. Bulgarevich et al.[15] used a trainable segmentation with a random forest classifier to segment ferrite, pearlite, and bainite in light optical micrographs of three-phase steels. Although methods for quantifying multi-phase microstructures have been suggested, the annotation and efficient segmentation of different microstructure constituents solely from LOM or SEM micrographs are not satisfactory.

As opposed to these works, supporting correlative electron backscatter diffraction (EBSD) information is used in the LOM and SEM annotation procedure to lay an appropriate foundation for learning. Moreover, the aforementioned conventional CV or ML approaches require complex image processing pipelines and elaborate feature engineering to render predictions robust against variances[14]. In contrast, the applied DL methods are directly based on input and target output image pairs (representation learning). Their application to microstructure recognition demonstrated the potential for quantitative microstructure analysis[16,17], steel type classification[16], crack path prediction[18], and micromechanical damage segmentation[19]. A CNN architecture referred to as U-Net[2] has proven its merit in the latter work and represents a common starting point due to its numerous implementations in different DL frameworks and image processing tools[20,21]. Therefore, this architecture represents a suitable candidate to derive best practices.

## Results

This work addresses the task of distinguishing lath-shaped-bainite regions (hereafter called foreground) from other phases such as polygonal and irregular ferrite with dispersed granular carbon-rich 2nd phase (background) in metallographic cross sections of complex phase steels. Therefore, the task is framed as binary segmentation which we address with supervised learning of CNNs. Unifying different methodologies such as specimen preparation, image acquisition, multimodal data registration, data fusion, deep learning modeling, and network visualization, all described in the "Methods" section, facilitates a holistic approach for microstructure inference. Ultimately, this puts us in a position to explore the interdependencies within and optimize this processing pipeline.

**Image sets and corresponding annotations**. Aside from LOM and SEM input micrographs, also their dense annotation masks (i.e., pixel-wise and binary labels indicating lath-shaped bainite) are crucial for supervised learning. For creating the annotation masks, additionally correlative EBSD data was used, see the section "Microscopy". The EBSD data gives access to accurate and highly quantitative features such as pixel-wise crystal orientation, nicely complementing the qualitative, mostly topography-sensitive information from our LOM or SEM data. Therefore, this analytical technique allows the accurate distinction of phases (and hence reliable dense annotations) even when facing complicated multi-phase microstructure scenarios. In the following, results of the LOM data preparation are shown. The SEM images were treated accordingly.

Figure 1 shows a LOM image (a), different overlays of LOM with suitable EBSD-derived characteristics (b–e), and the resulting annotations of lath-shaped bainite based on the LOM image and this EBSD-derived information (f). Enlarged details in (g–j) illustrate how unique grain color maps or grain boundary visualizations can be used to precisely define the boundaries of the lath-shaped bainite regions. For instance, it is visible which second phase particles belong to the object or are part of the surroundings (red encircled region in Fig. 1g–j). This data also helps

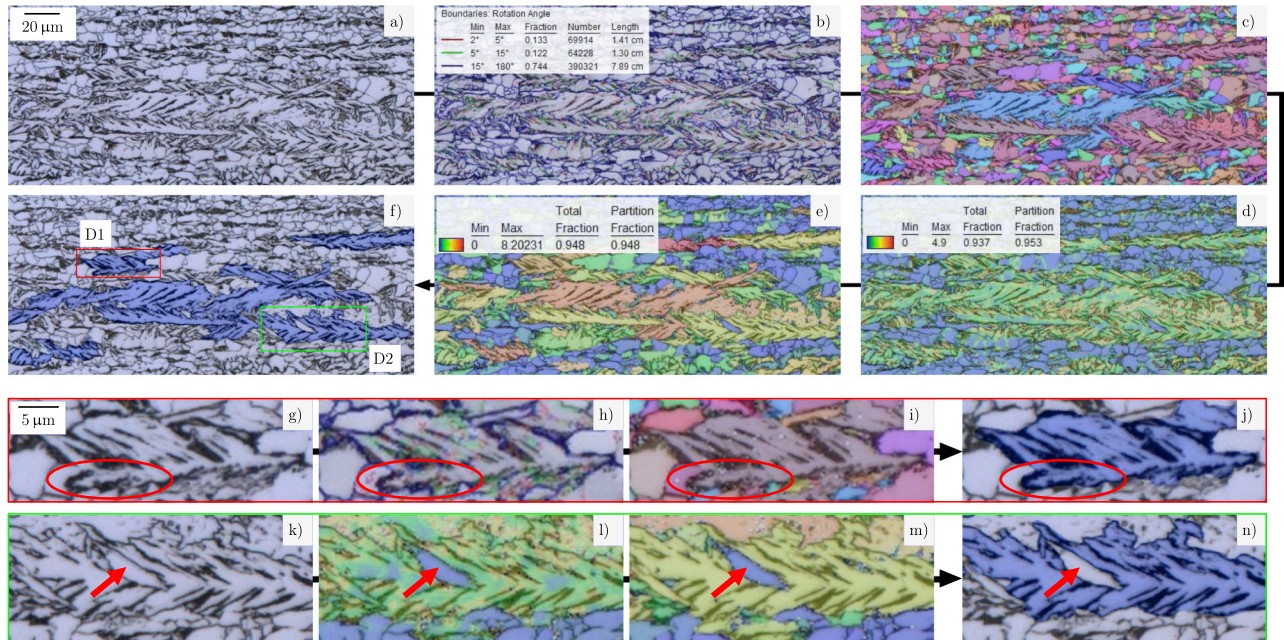

**Fig. 1 Illustration of correlative microscopy approach for objectively annotating lath-bainite regions. a** Original LOM micrograph. LOM overlayed with **b** an EBSD-derived grain boundary map, **c** unique grain color map, **d** kernel average misorientation (KAM) map, and **e** grain orientation spread (GOS) map. **f** LOM with annotated lath-bainite regions based on EBSD information. Detail views D1 (red frame) and D2 (green frame) are highlighted here. D1 in figures (**g–j**) displays how **h** grain boundary and **i** grain visualizations are used to correctly annotate the exact boundaries of the lath-bainite. In contrast, D2 in figures (**k–n**) demonstrates how **l** KAM and **m** GOS indicate polygonal ferrite grains in or adjacent to the lath-bainite region.

**Table 1 Intersection over Union metrics of U-Net-based networks trained on the light optical microscopy data set for different model initializations and downscaling factors.**

| # | Model | Model initialization | Downscaling factor | $IoU_{bg}$ | $IoU_{fg}$ |
|---|---|---|---|---|---|
| 1 | Vanilla U-Net | Random | native | 87.0 ± 1.5 | 70.2 ± 1.2 |
| 2 | Vanilla U-Net | Random | 0.5 × 0.5 | 87.2 ± 1.9 | 69.5 ± 1.8 |
| 2$^v$ | Vanilla U-Net | Random | 0.5 × 0.5* | 86.8 ± 1.5 | 69.8 ± 1.3 |
| 3 | U-Net VGG16 | Pre-trained | native | 87.6 ± 0.9 | 71.3 ± 1.7 |
| 3$^v$ | U-Net VGG16 | Random | native | 86.3 ± 1.3 | 69.3 ± 1.5 |
| 4 | U-Net VGG16 | Pre-trained | 0.5 × 0.5 | 87.1 ± 1.7 | 71.6 ± 1.7 |

The superscript $^v$ indicates a validation experiment conducted to test a particular hypothesis, as described in the section "Segmentation results". The superscript * indicates that downscaling was performed after tiling.

if grain boundaries are not clearly visible in LOM images due to weak contrasting. Additionally, when the determination of the class affiliation is impeded due to microstructural units with intermediate morphology between lath and granular, complementing EBSD information can provide a remedy. Without EBSD data, this assignment would have to be done solely based on the microstructure's visual appearance, which can lead to disagreement between human experts and inconsistencies during annotation. Enlarged details in k–n show that misorientation parameters, and specifically low values (blue color) for kernel average misorientation (KAM) or grain orientation spread (GOS) in Fig. 1l + m, indicate polygonal ferrite grains adjacent to or even inside lath-shaped regions (red arrows in Fig. 1k–n). These embedded or adjacent polygonal ferrite grains should be excluded during annotation. The distinct crystallographic orientation of the embedded grain (see Fig. 1c) does not suffice to unambiguously exclude it from the lath-bainite class. However, intragranular misorientation metrics can characterize such marginal cases as ferritic regions since small intragranular misorientation is incompatible with the notion of lath-shaped bainite[22].

These illustrations also clearly show the difficulty of the segmentation task at hand because the different phases are not

distinguishable by gray value distribution, show very complex-shaped borders, and can exhibit objects of one class inside objects of the other class.

**Segmentation results.** Two CNN architectures, a vanilla U-Net and VGG16 U-Net, were trained according to the section "Deep learning methodology" with LOM or SEM tiles (image crops) along with their corresponding dense annotations. These tile images were either provided at native scale or downscaled. The segmentation performance is evaluated in terms of the intersection over union ($IoU$) metric for the foreground (fg: lath-shaped bainite regions) and background (bg: polygonal and irregular ferrite with dispersed granular carbon-rich 2nd phase) as positive classes each. The given metrics for each model represent the average and standard deviation over five cross-validation trials and were typically evaluated on tiles. Since the aleatoric uncertainty component[23] introduced during training was previously confirmed to be negligible, the standard deviations given in Tables 1 and 2 are predominantly attributed to the k-fold sampling from the low-quantity data sets. This is shown in a diagram in the Supplemental, where the class-averaged $IoU$ is plotted over the fold number for two models. For this reason, and since the

**Table 2 Intersection over Union metrics of U-Net-based networks trained on the scanning electron microscopy data set for different model initializations and downscaling factors.**

| # | Model | Model initialization | Downscaling factor | $IoU_{bg}$ | $IoU_{fg}$ |
|---|-------|---------------------|-------------------|------------|------------|
| 5 | Vanilla U-Net | Random | Native | 69.2 ± 2.1 | 77.5 ± 2.4 |
| 6 | Vanilla U-Net | Random | 0.5 × 0.5 | 75.4 ± 2.9 | 77.9 ± 3.3 |
| 6$^v$ | Vanilla U-Net | Random | 0.5 × 0.5$^*$ | 72.7 ± 2.3 | 79.9 ± 2.8 |
| 7 | U-Net VGG16 | Pre-trained | Native | 71.0 ± 2.5 | 80.1 ± 2.4 |
| 8 | U-Net VGG16 | Pre-trained | 0.5 × 0.5 | 77.7 ± 3.2 | 80.4 ± 2.6 |
| 8$^v$ | U-Net VGG16 | Random | 0.5 × 0.5 | 68.4 ± 4.0 | 73.1 ± 1.5 |

The superscript $^v$ indicates a validation experiment conducted to test a particular hypothesis, as described in the section "Segmentation results". The superscript $^*$ indicates that downscaling was performed after tiling.

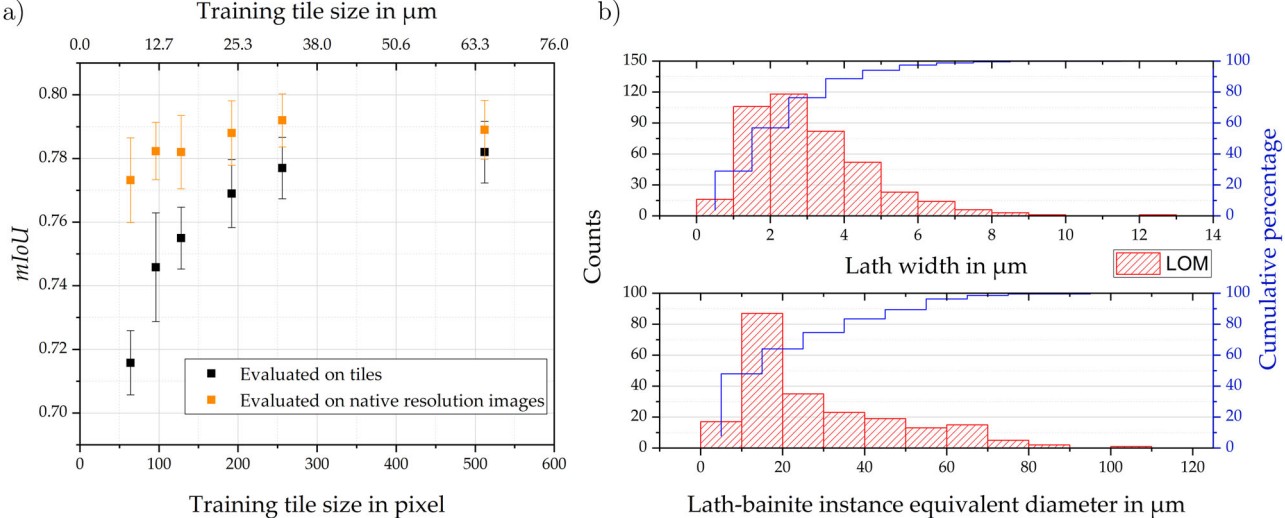

**Fig. 2 Influence of tile size on the network performance under consideration of microstructure feature sizes. a** Influence of tile size during training on the mean intersection over union metric (averaged over both classes) for the light optical microscopy domain and the vanilla U-Net model. Evaluation on tiles with a size equivalent to the training tile size (black) and evaluation on native resolution images was performed. Each data point and the error bar corresponds to the average and standard deviation over five folds. **b** Lath width and lath-bainite instance equivalent diameter histograms including a cumulative percentage for LOM.

same overall data was used to train the models, the average values can be utilized to deduce tendencies between the models within each modality.

In Table 1 the LOM image-trained models along with model initialization type, image resizing factor, and $IoU$ metrics are listed.

The model performances of the two architectures trained on native and downscaled image tiles (#1, #2, #3, and #4) correspond to accuracies around 90%. This is comparable to the discrepancy in annotation by human experts when relying solely on topography information. For the VGG16 U-Net architecture typically training from a pre-trained configuration was conducted, while the vanilla U-Net was trained from random weight initialization at all times. To facilitate comparability between the VGG16 and vanilla U-Net architectures, a random initialized VGG16 validation model #3$^v$ was introduced. It can be observed that for random initialization, the conventional U-Net scored slightly better than the VGG16-based variant (cf. model #1 and #3$^v$). At the same time, model #3$^v$ acted as the randomly initialized equivalent of the ImageNet pre-trained U-Net VGG16 (#3), therefore enabling assessment of the pre-training dependency. Their comparison showed almost 2% $IoU$ improvement utilizing pre-trained models. This indicates that even pre-training with domain-external data sets such as ImageNet can benefit material scientific tasks.

The downscaling of the images, in general, was performed before tiling, resulting in fewer tile images with sampling-induced

and interpolation-induced information loss but comparatively more context in individual tiles. Downscaling for the vanilla U-Net did not affect LOM image segmentation performance significantly. Similarly, this applies to the another experiment 2$^v$, where downscaling after tiling was used for validation purposes. Further, a higher scatter in $IoU$ is observed for models when downscaling before tiling is applied.

In the case of the vanilla U-Net, a foreground weighing factor $\alpha = 1.5$ to correct for the material-inherent class (i.e., phase) distribution imbalance was found during the hyperparameter optimization to improve the overall $IoU$ slightly. Introducing the same factor and weighted binary cross-entropy loss in the VGG16 variant did not change its performance in a statistically relevant manner.

As an additional ablation study, the impact of LOM tile size was tested on the random initialized vanilla U-Net networks. In this case, rather than introducing scaling factors, tiles were cropped with different sizes. The objective was to examine the relations between tile sizes and characteristic microstructural unit sizes. This study is exempt from the general procedure since model evaluation additionally took place on native resolution images rather than exclusively on tiles. In Fig. 2, the results of this study are summarized.

This graph shows that evaluation on tiles is strictly detrimental compared to evaluation on native resolution images (1024 × 1024)

for the segmentation of such micrographs. For small tiles, the performance gap between evaluation on tiles and native resolution images amounts to 6% *IoU*. Both curves converge for tile sizes approaching the native resolution. The orange curve represents the effect of tile size during training and indicates that even when tiles of $64 \times 64$ pixels or $8.1 \times 8.1\ \mu m^2$ are utilized, training is not hampered substantially (less than 2% *IoU* decrease).

Moreover, in a preliminary study, we investigated the influence of additional uncertain tiles, i.e. tiles with structures that somewhat resemble laths but are distinct. In this case, additional debatable structures did not affect the performance to a significant extent.

For the SEM image-trained models (Table 2), the difference between foreground and background *IoUs* is less pronounced than for the LOM. Moreover, downscaling before tiling impacts the $IoU_{bg}$ strongly, where downscaling proves advantageous (cf. $IoU_{bg}$ of models #5 and #6, or #7 and #8). Analogous to the LOM

case, the influence of downscaling after tiling was investigated in a validation experiment #6$^v$, showing comparatively more uniform improvement of about 3% *IoU* across both classes (cf. #5 to #6$^v$). While the result of downscaling improving performance seems counterintuitive at first glance, its plausibility will be validated later. The pre-trained U-Net VGG16 trained on downscaled data (#8) achieved the best IoU for SEM images.

However, while pre-training contributed to only mediocre improvements for LOM (cf. model #3 and #3$^v$), it resulted in a significant foreground and background *IoU* improvement for SEM, of approximately 8% *IoU* (cf. model #8 and #8$^v$). Further, the SEM model performances confirm the LOM-case observation that the vanilla U-Net performs better for random initialization. The performance of best model #8 corresponds to 88.4% accuracy.

In Fig. 3, the resulting segmentation map predictions of the best vanilla U-Net (Fig. 3a+b) as well as best random initialized

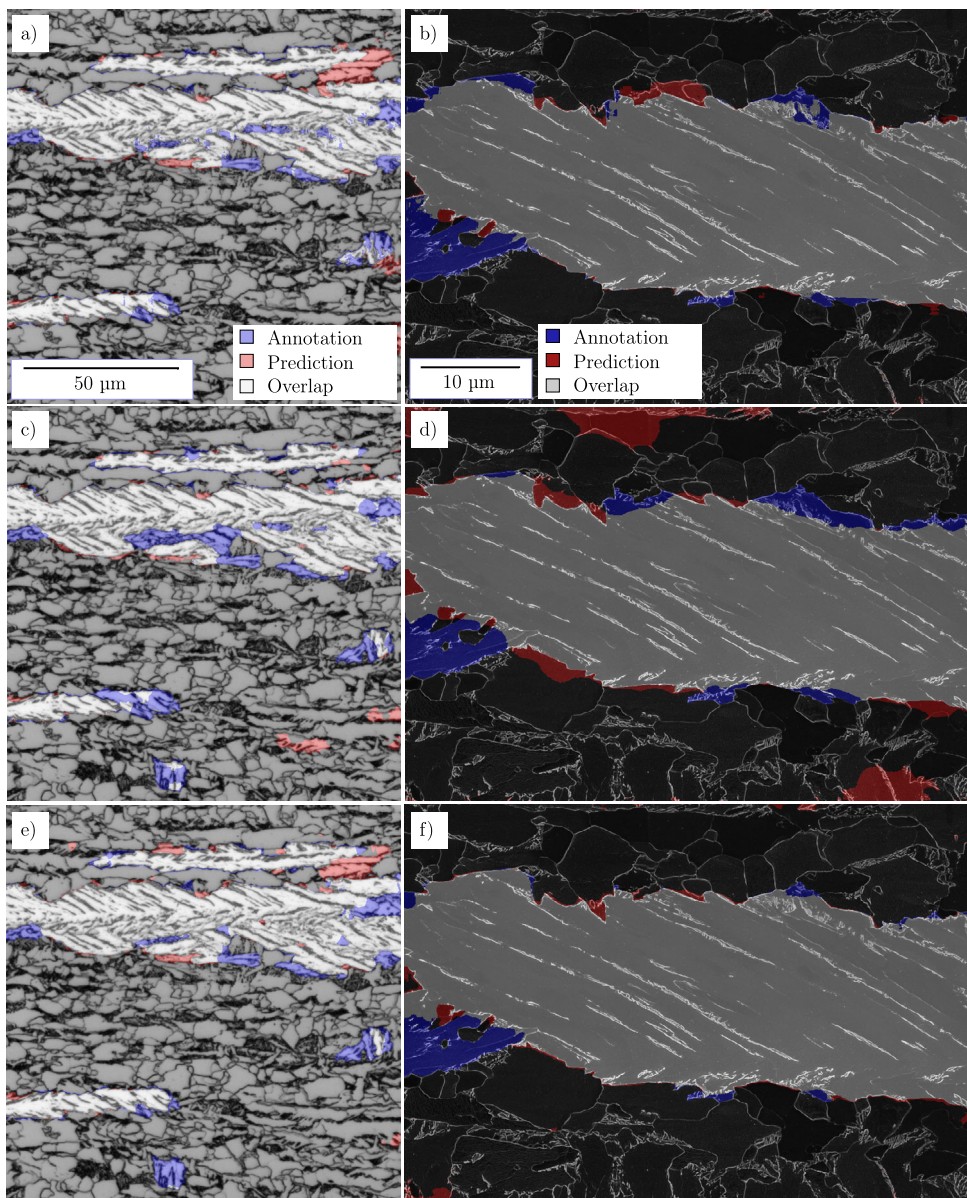

**Fig. 3 Light optical and scanning electron micrographs superimposed with lath-bainite predictions of different models and annotated regions showing the comparison between model prediction (red) and manual expert annotation (blue). a, b** Random initialized vanilla U-Net (model #1 and #6). **c, d** Random initialized U-Net VGG16 (model #3$^v$ and #8$^v$). **e, f** Pre-trained U-Net VGG16 (model #3 and #8).

and pre-trained U-Net VGG16 (Fig. 3c, d and e, f) models are compared to the annotations for both modalities. Note that the illustrated images are full-frame, while for training and testing, tiles of such images were used. The vast majority of lath-shaped bainite regions are well identified, and the predictions are widely in accordance for all three models of each modality. Moreover, locations of erroneous predictions match in the models to a large extent. Under-prediction (blue) occurs at individual smaller foreground objects or at the borders of extensive lath-shaped bainite regions, where annotated parts do not exhibit a clear lath structure. Over-prediction (red) tends to arise mostly in smaller areas in which carbides or grain boundaries resemble lath shapes. In both modalities, but especially in the SEM-trained case, the random initialized VGG16 (c, d) falls short as opposed to the other models, which is mirrored by the performance metrics in Tables 1 and 2.

## Discussion

To achieve reliable and objective microstructure inference, an understanding of fabrication, microscopy, DL methodology, and their interdependencies is required. It is important not merely to look at the images and corresponding annotations as an isolated step but also to consider building a DL model as a holistic approach where specimen preparation, reproducible specimen contrasting, and suitable image acquisition techniques are of tremendous importance[9].

In our study, we successfully trained both random initialized networks and pre-trained networks with comparatively small data sets of approximately 50 and 30 images for LOM and SEM, respectively. Ascribing to the reproducible specimen preparation and imaging (i.e., low variance in the metallography process), we believe the observed scatter in the data to be predominantly materials' microstructure-based. Under this assumption and considering the characteristic sizes of microstructural features (such as lath width and grain sizes, see Fig. 2), it is likely that the imaged area is largely representative of the microstructural scatter. For such datasets generated through reproducible processing, this invalidates the frequent preconception of DL being only applicable for large-scale data sets.

Special attention was paid to specimen preparation, optimal contrasting during etching, and consistent settings during image acquisition. Moreover, the very reproducible imaging settings, e.g., viewing perspective, brightness, and contrast, lead to a low degree of material-extrinsic variance in the images compared to real-world scenario image sets. The pronounced planarity of the metallographic cross-sections avoids geometry-related image shading and distortions. Class imbalances often pose a challenge for learning. Due to the comparatively lower magnifications during image acquisition, the LOM image data set is representative of the microstructure in terms of phase fractions, where lath-shaped bainite is a minority class (28%). This poses a material-inherent class imbalance. On the other hand, the lath bainite phase was oversampled during SEM image acquisition to correct for the imbalance. If such imbalances were not artificially corrected at the image acquisition stage, post-processing techniques such as sampling or weighting methods could be applied to account for them.

The choice of imaging modality primarily depends on the scale on which relevant microstructural features are to be expected. For instance, while LOM might be well suited to deduce lath-shaped regions in CP steels, SEM was incorporated as it additionally contains information on the exact nature of carbon-rich second phases, which renders the distinction between bainite subclasses possible. Perspectively, when the data quantities and imbalances between both modalities are matched, the more suitable modality for lath-shaped bainite prediction or other tasks can be concluded.

Assigning the ground truth, i.e., correctly annotating the lath-shaped bainitic regions, is challenging, and disagreements between human experts can arise when purely relying on the microstructure's visual appearance in LOM or SEM. By incorporating correlative EBSD data, even though for just a part of the image set, the objectivity and reproducibility for annotating micrographs are improved. However, annotating the foreground regions manually by tracing their perceived object boundaries on a tablet can still lead to some inconsistencies.

All segmentation models achieve performances that are comparable to expert segmentations performed in the absence of EBSD data. Since two research groups applied their DL best practices and the architectures are fairly similar, the changes in performance are not extremely pronounced. Nonetheless, for the first time applying DL best practices of two groups on identical materials data gives important insights. These insights are essential to facilitate subsequent major improvements through adapted pre-processing, architectural choices, and training procedures. The similar segmentation results of U-Net-based models point towards general performance robustness concerning different architectures and training strategies. Namely, no severe performance decrease was observed by different initial network conditions (random initialization as opposed to ImageNet pre-trained model), different internal padding, and normalization strategies. The regions where the models fail are regions where human experts would primarily make mistakes during manual segmentation. Especially, this applies to instance grain boundary regions with the absence of distinct lath structures, which are prone to low inter-rater reliability.

The segmentation enables the accurate calculation of lath-shaped bainite phase fractions. Reported *IoUs* for the foreground (lath-shaped bainite, Tables 1 and 2) correspond to minor phase fraction errors in the range of 1% compared to human expert annotation. This error is lower than the variance in manual human expert evaluation. These are remarkable results considering the intricacy of the segmentation task at hand. A prerequisite to achieve accurate phase fraction predictions is that the training data is not significantly skewed towards a specific class. Skewed data sets would result in models that favor the majority class[24]. Therefore, for accurate phase fraction estimation of relatively uncommon phases, imbalance correction is advised.

Since there are some deviations along the border of lath-shaped bainite objects, localization of phase boundaries is only possible to a limited extent. These border deviations are potentially partly attributed to the aforementioned border annotation inconsistencies and hamper the calculation of individual bainite object morphological parameters associated with the objects' spatial extent. On the other hand, segmentation enables the analysis of inner morphology of specific phases in the first place. In this case of CP steel microstructures, it facilitates the detached calculation of the lath-shaped bainite regions' lath characteristics (e.g., lath-width) instead of calculating these characteristics for the whole image, yielding a more differentiated and sophisticated microstructure quantification. These morphological parameters are known to impact mechanical properties significantly[25,26]. Furthermore, the also accessible relative spatial distribution of phases in such heterogeneous microstructures affects local fatigue properties severely. Such focused microstructure analyses are the prerequisite to establish processing-microstructure-property correlations. Furthermore, reliable and high-fidelity segmentation has implications for automated and targeted microscopy.

The isolated tile context influence was demonstrated in Fig. 2a by cropping tiles with different resolutions from the native LOM images and using them for training while evaluating either on tiles of the same size or native resolution images. While the orange data points (evaluation on full-frame as-acquired images)

entail only information on the tile size influence at training time, the black data points (evaluation on tiles) additionally encompass information on tile size at test time. The results indicate that the context contained in the tiles or images provided during training and testing is of pronounced importance. Both data point series converge, while evaluation on native resolution images is always advantageous as image edge effects are suppressed. Therefore, while at training time, tiling is often inevitable due to GPU memory constraints, during testing, where the image size is not restricting, avoiding tiling proves beneficial.

In order to compare tile sizes with characteristic microstructural entity sizes, the lath width histogram (top) and one for the equivalent diameter of lath-bainite regions (bottom) are provided for the LOM modality in Fig. 2b. As the training tile size approaches the lath width, a detrimental impact on learning is observed. However, the fact that even learning with reduced $8.1 \times 8.1\ \mu m^2$ tile sizes and evaluating on whole images shows a drop of 2% $IoU$ only, indicates that either a few parallel carbide films at lath boundaries suffice as features or that the network adapts to focus more on less prominent short-range features. Nonetheless, in this lath-bainite segmentation case, where parallel but distant inter-lath carbide islands are relevant to deduce the foreground, it is valuable to avoid a small training tile field of view to increase the likelihood of obtaining multiple parallel carbide clusters within a single image. For the evaluation on tiles (black data series in Fig. 2a), it can be observed that reducing the tile size is even detrimental at larger physical tile sizes. This is associated with image edge effects depending not only on feature separation (i.e., lath width) but also strongly on instance size. Since individual lath-bainite instance diameters range up to more than 100 μm, such image edge effects play a role also at larger tile sizes. Both lath width histograms for LOM and SEM (not shown) exhibit similar distributions, with LOM lath widths' shifted slightly towards larger values, presumably due to the different etchings and worse optical resolving power. The reasoning applied here can be carried over to many microstructure inference tasks since objects of interest in metallographic micrographs (e.g., phases) and features within these objects are often comparatively more dispersed than in many real-world scenario images.

Based on these results of the tiling study, it is comprehensible that the context increase, associated with downscaling before $256 \times 256$ pixel tiling, would not have a major impact on LOM performance as suggested in Table 1. This is plausible given the large physical pixel size in native scale LOM images, which results in tiles that already capture sufficient microstructural features. However, assuming the trend in Fig. 2a being modality-independent raises the question of whether SEM results would benefit from more image context at the learning and testing stage, especially considering that SEM tiles without downscaling have a physical tile size of ~14.2 μm.

It is reasonable to assume that such SEM tiles do not represent the lath structure appropriately but only contain fragments of the lath-shaped regions, thus impeding learning. This question, amongst others, was addressed by SEM downscaling experiments, where downscaling before tiling resulted in a $IoU_{bg}$ performance improvement (cf. model #5 and #6, or #7 and #8). The reason for $IoU_{bg}$ improving, in particular, can be traced back to the dataset acquisition and pre-processing stages. Namely, this can be attributed to the lath-shaped bainite oversampling (localized in the image centers) combined with additional mirror padding in the downscaled dataset to still extract $2 \times 2$ tiles. This combination skews the downscaled SEM dataset towards the background class. When applying downscaling before tiling, aside from this SEM dataset-specific change in class distribution, image context, as well as context provided to the so-called network receptive field, is increased artificially, and

information loss is introduced. In order to discern these influence factors' contributions to the performance, an additional validation experiment (#6$^v$) was performed, where tiles used for model #5 were downscaled as an exception (i.e., downscaling after tiling). In this case, the tile context increase and aforementioned mirror padding differences, both with respect to model #5, are absent. Despite the still present information loss in experiment #6$^v$, the downscaling after tiling leads to a relatively uniform and notable improvement of 3% $IoU$ for both classes. This can be ascribed to the fact that by downsampling more physical area is taken into consideration at each network layer. In contrast, when downscaling is discarded, only later layers can consider sufficient context when extracting features due to the small physical pixel size in the SEM images.

Therefore, aside from ensuring appropriate feature representation in images, it is important to select a network architecture for the task at hand that takes a sufficient amount of context into consideration. Characteristic image length scales (e.g., phase boundary pixel distance) change depending on the applied magnifications and image resolutions required to resolve relevant features during image acquisition. In such cases, it can be beneficial to adapt the image region that the network considers, also referred to as theoretical receptive field (TRF), accordingly. In Luo et al.[27], the effective receptive field (ERF) metric was proposed for CNNs and was empirically computed for several architectures. The ERF revolves around the notion that not every region within the TRF is taken into account equally. In fact, their predicted ERFs were substantially smaller than the TRF and showed a 2D Gaussian distribution that strongly decays towards border regions of the TRF. This means, the closer a pixel is to a target pixel, the more it influences the target pixels' predicted class[27]. This represents a CNN-based inductive bias appropriate for many scientific segmentation challenges, such as for fatigue damage localization where image features are dense[19]. However, for phase segmentation tasks, where long-range features (parallelism of distant carbide islands) are relevant, more general Attention-based networks[28] could improve segmentation performance in the future. The observation that the scale of features determines the optimal downscaling factor has led to specialized architectures. Especially in such multi-class segmentation or classification tasks where features are distributed across scales, aggregation of distinctly dilated convolutions is reasonable[29]. In conclusion, it is important that individual tile images comprise sufficient learnable feature information, and the architecture facilitates their appropriate extraction and processing.

Downscaling, if this condition is fulfilled, but especially when severe information loss is expected as its consequence, should be avoided. In our case, while segregated carbides at lath interfaces are slender, we believe that these features are still largely preserved when $0.5 \times$ downscaling is applied. This is in accordance with[30], where a plateau of nearly constant performance for a range of downscaling factors ($0.2–0.5 \times$) was demonstrated. Downscaling factors below a threshold are typically accompanied by a significant information loss and a decrease in segmentation performance. Such information loss can be ascribed to image downsampling and non-ideal interpolation. Literature[30,31] suggests that this threshold depends on the specific foreground class. In these works, it was shown that specific classes that have fine features or small object extent profit from discarding downscaling operations.

To summarize, in phase quantification, where long-range features such as the morphology of grain boundary traces are relevant, downscaling before tiling can potentially improve performance and accelerates training. When applying downscaling, it should be used consistently at training and testing time. This holds especially true if no preventive measures such as scaling data augmentations are taken at training time.

Considering random initialized models, the vanilla U-Net scoring better than the U-Net VGG16 for both modalities can be hypothesized to be ascribed to the following factors:

- The learnable transposed convolution upsampling can recover spatial localization in the decoder more accurately[32].
- Batch normalization applied in the encoder of the vanilla U-Net can improve performance[33].
- In vanilla U-Net training, the imbalance in the data set was corrected by employing a class weight inside the focal loss function.

The distinct loss function was experimentally invalidated to be the cause for the performance difference in architectures. The other individual factors' influence was not systematically retraced.

The fact that pre-training led only to a minor improvement (almost 2% $IoU$) for the U-Net VGG16 in the LOM case leads us to conclude that the 50 full-frame LOM images with relatively even class distribution suffice for training such a binary segmentation model. While this is dependent on the exact problem and model to be trained, we infer that given such data, U-Net-based models, which score satisfactory results at such ambitious phase segmentation tasks, can be trained from scratch. In contrast, for the SEM case, the pre-training culminates in an $IoU$ increase of 7–9% over the random initialized U-Net VGG16 (cf. model #8 and #8$^v$). Therefore, the pre-training dependence is comparatively more pronounced in the SEM modality. This can be ascribed to the fewer amount and higher magnifications of SEM images, hence covering considerably fewer distinct microstructure scenarios. Therefore, when the data set comprises few images covering a small field of view, we advise pre-training with readily available data sets. While ImageNet encompasses a wide range of classes, the noise characteristics and image formation in microscopic images are different. Potentially, pre-training with other data sets exhibiting a smaller domain gap such as miscellaneous nanoscientific objects in SEM[34] or ultra-high carbon steel phases SEM[35] can be advantageous.

The random k-fold sampling of low-quantity data, especially in SEM, results in notable $IoU$ scatter. In such cases, stratified sampling and training can prove beneficial before the deployment of the model. The reproducible preparation, mounting, and imaging (i.e., low intra-domain variance) rendered the data augmentation, and corresponding hyperparameter tuning negligible as the performance improvement associated with it for both modalities was minimal. This implies that data augmentation is not generally essential for small-scale data sets, but only when the applied transformations render the training set more representative of the test set. In instances where such material-extrinsic variance can be ensured to be insignificant, data augmentation through simple spatial (affine and even elastic) or intensity transformations can be evaded. Therefore, such models trained on comparatively small data sets are suitable for tasks with inherently small scatter, such as quality inspection, where recurring tasks and predefined workflows are set. When, for instance, etching-based contrasting methodologies are concerned, reproducibility can be difficult to attain.

To this end, a generalization study was conducted to test the transferability of this model, trained with low variance data, to an alternate data domain. Therefore, the previously best SEM vanilla U-Net model #6 was tested on an SEM image of a surface etched with Nital as an exception. In contrast, another model was trained with dedicated augmentation settings to improve the performance on the alternate etching domain. Figure 4a + b illustrates the comparison of the source domain (electrolytic etch) with the alternate target domain. Moreover, Fig. 4c + d depicts the segmentation of Fig. 4b using model #6 and a model trained with solely brightness and contrast augmented images of the source domain. The degree of both augmentations was optimized for the target domain.

A substantial improvement of $IoU_{bg}$ and $IoU_{fg}$ of 65% → 69% and 35% → 54% is achieved, respectively. This corresponds to a change in phase fraction prediction of 17% → 33% for a manually labeled phase fraction of 44%. Evidently, the segmentation in Fig. 4d is not satisfactory since applied augmentations do not close the domain gap entirely. More elaborate image transformations would be required to align the domains since the secondary electron image formation is strongly affected by the different topographies. Nonetheless, the fact that even simple targeted optimizations of low-variance training data can cause such improvements implies that dedicated data augmentation pipelines can presumably render models robust against a large range of perturbations in the specimen preparation or imaging. For instance, in our prior study[19] a substantial improvement was achieved by augmentation of our high-variance data. When training images are acquired from different instruments or at different institutions, such regularization methods become increasingly relevant. In such instances, it is essential to track and store all relevant process parameters along the entire process chain in a structured and ideally semantics-informed database. Moreover, this outcome foreshadows that advanced augmentation with generative adversarial networks (GANs)[36] or domain adaptation[37], potentially can address even more challenging generalization demands of the materials science community in the future.

Interpretability and explainability of DL models are important to build trust and push for successful implementation in day-to-day applications. Moreover, it can help in finding failure modes of models and give insights on tackling them. To that end, we computed network visualizations introduced in the section "Deep learning methodology" that highlight regions or concepts within an image that affected the network's decision. Meaningful examples of Grad-CAM and NetDissect visualization from several network layers are illustrated in Figs. 5 and 6, respectively. Furthermore, Grad-CAM visualizations of all network layers for both architectures can be found in the Supplemental.

Grad-CAM masks are generated for a particular convolutional layer of the trained models #1 and #3 for a specific class. These masks are formed by a weighted average of all activation maps originating from all the target layer filters. Hence, the masks highlight those regions in the input image that the specific layer treats as essential for predicting the specified class. Thus, by looking at the Grad-CAM masks of various layers for lath-bainite and background classes, we can deduce how the trained model predicts a segmentation mask for a given input.

Although the following observations are qualitative, they are very helpful in interpretation. Concerning the background in the LOM segmentation, strong activations are caused to a certain extent by particles of the carbon-rich 2nd phase (b, c, e, h) and for the most part by grains and grain boundaries of the polygonal ferrite (b, d, f, g, i). Moreover, in the down4convr1 layer (c), there is a focus on grain boundary junctions, such as triple and quadruple points, which are discriminative features. Activations in the vanilla U-Net and VGG16 U-Net mostly match (note that e and h show similar activations that differ in scaling and the degree of focus on carbon-rich clusters). However, towards the end of the decoder (layer up4convr2), the vanilla U-Net focuses on polygonal ferrite grain boundaries (f) to determine the final output, whereas the VGG16 U-Net focuses on the grains themselves (i). The model during decision-making puts emphasis on image features that correlate with how human experts interpret the image. For instance, the decision for the background

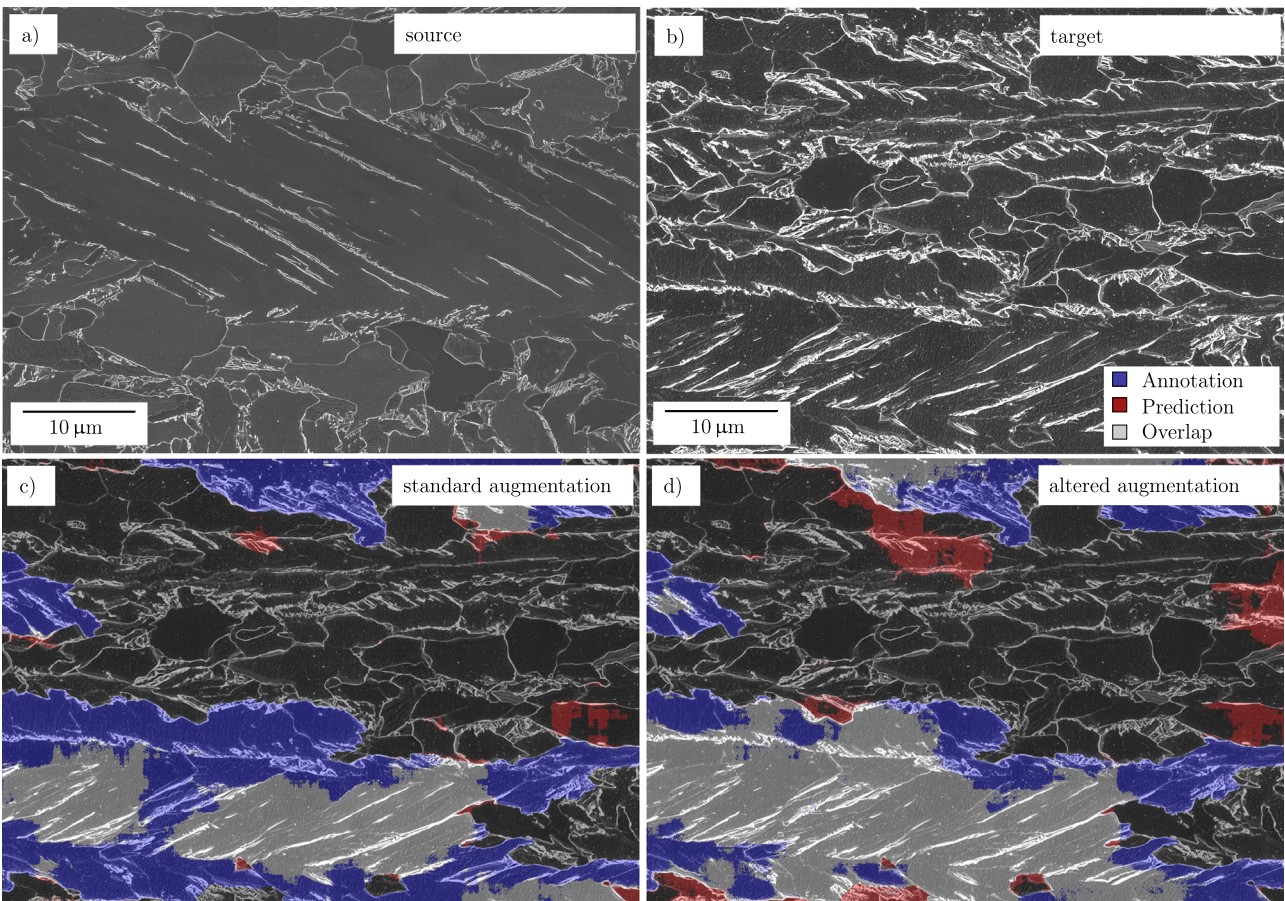

**Fig. 4 A generalization study for alternatively etched surfaces. a+b** Comparison of the electrolytically-etched source image domain (**a**) and the alternate Nital-etched specimen (**b**). **c** Segmentation results of model #6. In this case, augmentation parameters were chosen for the source domain (**a**).
**d** Segmentation results of a model like #6 but with modified brightness and contrast augmentation to improve performance on the alternately etched (target) domain. The legend for (**c**) and (**d**) as well as the micron bar for (**b**), (**c**), and (**d**) is positioned in (**b**) to avoid concealing relevant regions in the segmentation results.

will mostly depend on the existence of ferrite grains, which are comparatively more equiaxed. It should be noted that the vertical and horizontal line artifacts visible in Fig. 5d are presumably attributed to the checkerboard problem associated with transposed convolutions[38] as such artifacts do not occur in the VGG16 case that used bilinear upsampling.

Lath-bainite activations in some layers are induced by second phase particles and grain boundaries in general or by elongated second phase particles and grain boundaries. However, the strongest reactions are caused by pronounced, more extensive lath regions. An analogy to the human expert examination can be supposed here, too. Pronounced, more extensive lath areas should also be noticed strongly by the human eye because of regular lath structure compared to the surrounding. Significant differences in feature importance between the different U-Net architectures were not found.

In contrast to Grad-CAM, NetDissect enables us to analyze what different filters in the model extracted from an input image, regardless of its contribution to the final segmentation map. This technique offers us the prospect of finding disentangled feature extractors from the model, which make sense to a human expert, see Fig. 6. Note that these exemplary images represent only a small portion of filters utilized in the whole network. In the case of LOM images, relevant features include 2nd phase particles plus elongated grain boundaries (a), lath-shaped 2nd phase particles (b), and the area of more extensive grains (c). Thus, an analogy to human expert interpretation can be assumed here as well.

Moreover, considering that lath-shaped 2nd phase particles are relevant features, similarities to how feature engineering is performed during conventional ML or CV can be seen, too. For instance, in Müller et al.[10] a sliding window technique that utilizes a Prewitt[39] edge detection filter to calculate directionalities of the 2nd phase particles is applied. Directionalities are used, in combination with a neighborhood analysis, to detect lath-shaped regions.

Given the large number of different materials and processes, and the time-intensive generation of data sets for many tasks, materials science will always be accompanied by data scarcity. It is all the more important that strategies of model generalization to alternate materials or processing conditions are pioneered. As a consequence of emerging high-speed image acquisition technologies, annotation processes often pose the bottleneck in the creation of statistical data sets. This particularly holds true for the supervised learning of segmentation models in the material scientific domain. By pushing the correlative approach with EBSD measurements forward, routines for automatically generating annotations based on EBSD data can be developed. This promises to improve the annotation quality and make it less labor-intensive. Moreover, it enables further segmentation tasks to be addressed, e.g., segmenting lath-shaped and granular bainite as well as distinguishing bainitic and pro-eutectoid ferrite.

Nonetheless, generalizing data-driven methodologies and alternate learning strategies will be indispensable to cope with material diversity. In literature, different training strategies to

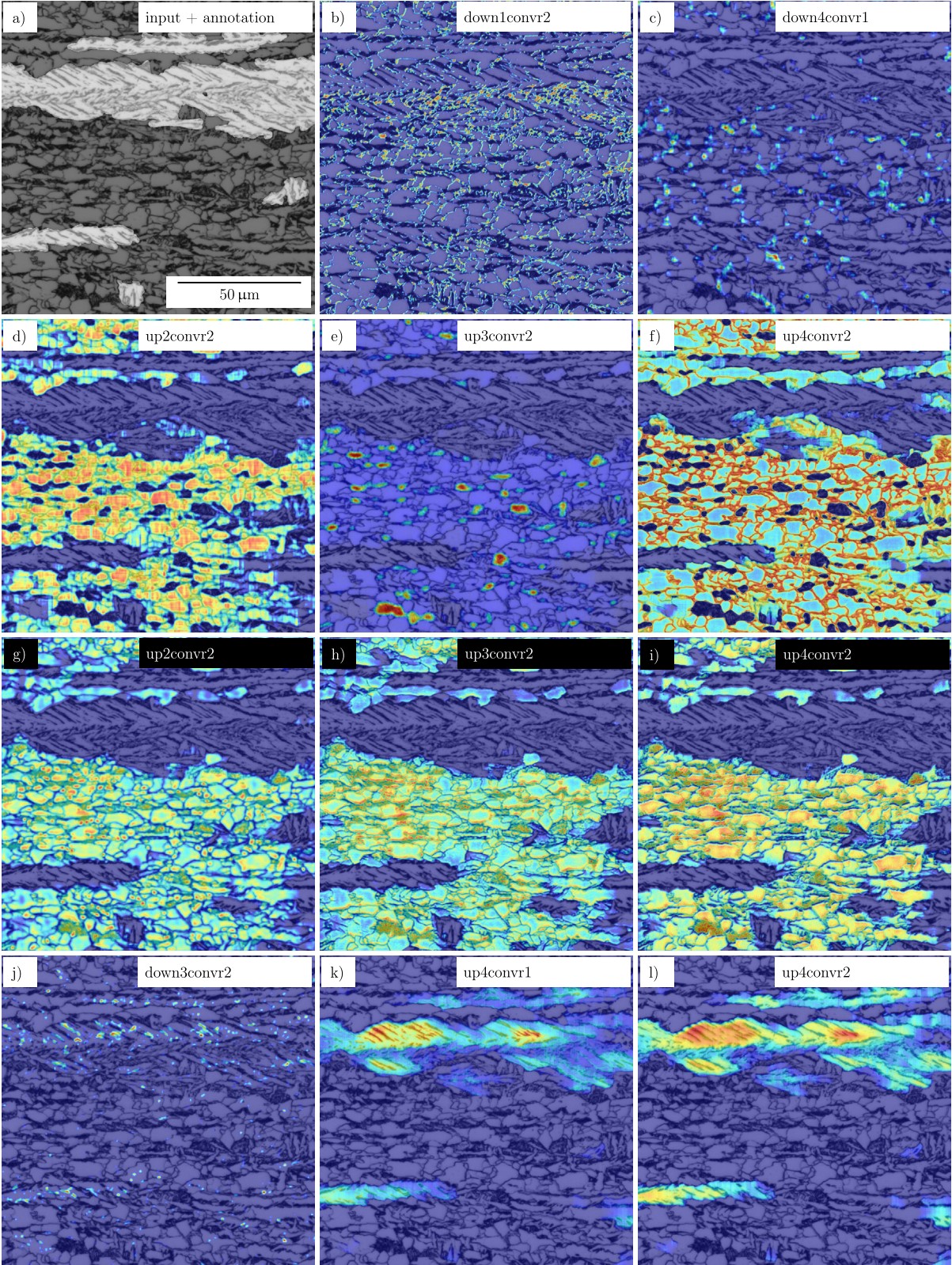

**Fig. 5 Gradient-weighted class activation maps indicating image regions that dictated the decision of the network.** The red color in (**a**–**i**) and (**j**–**l**) highlights regions that incentivize the model to vote for the background class and lath-bainite, respectively. The Grad-CAM maps are computed for specific layers (see panel legends) considering light optical microscopy experiments #1 (black font) and #3 (white font). Layer names are indicated in Supplementary Fig. 1.

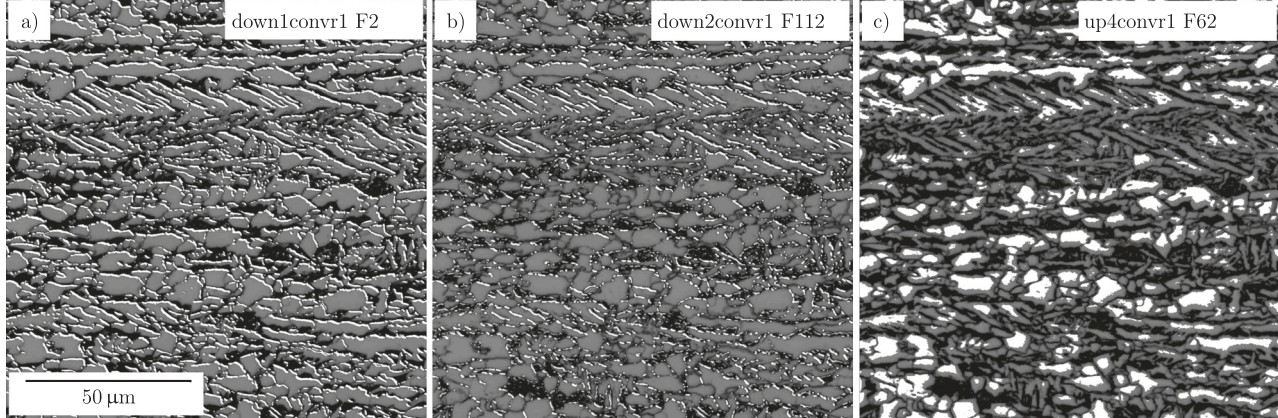

**Fig. 6 Thresholded activation maps of specific convolution filters using the NetDissect method. a–c** Illustrate different relevant extracted features. The high gray value regions indicate disentangled concepts that were learned in experiment #1. Panel legends state the layer, designated in Supplementary Fig. 1, and filter numbers (FX) of feature extraction.

tackle the sparsity of annotated data have been developed which rely on comparatively less data. These can be adopted for the segmentation of metallographic phases or the materials community in general.

Rather than providing pixel-wise annotations for training a segmentation network, in a weakly-supervised learning setting, e.g., image-wise annotations are used. There are different annotation abstraction levels ranging from bounding boxes[40] to naming the classes present in an image[41]. Typical methodologies rely on classification networks which provide seeds for the segmentation network, and constrained seed region growing to respect object boundaries[41,42]. In recent years a leap in weakly-supervised segmentation performance was achieved[43], rendering it a promising method for phase fractions. This is affirmed, since well contrasted grain boundaries presumably can pose distinct and suitable borders for region growing. In particular, for metallographic segmentation tasks in which target phases are often dispersed across the whole image, pixel-wise annotation is cumbersome. Here it can be particularly worthwhile to replace manual pixel-wise annotations by appropriate weak labels.

Alternate techniques called semi- or unsupervised domain adaption evolve around the idea that for a specific task (e.g., segmentation) annotated data of one source domain (e.g., material A) can be used together with non-annotated or minimally annotated data of a target domain (e.g., material B) to produce meaningful predictions in latter. The methods achieving this rely on feature matching between both domains, self-training to provide pseudo labels or generative networks to produce target data[37]. The range of materials and processes that can be covered with such techniques in material scientific challenges is yet to be unveiled. Moreover, the materials science domain can profit from its longstanding experience in knowledge-driven, realistic simulation techniques such as phase field simulations. The resulting synthetic data can be exploited in domain adaptation to obtain annotated data in a source domain or for pre-training[44].

Another promising candidate to reduce annotated data requirements are physics-constrained DL models[45]. Rather than supplying a multitude of input-output pairs, conditions that represent domain knowledge are imposed on the output space. In such cases the domain knowledge is typically encoded into the loss function. For microstructure inference, laws from thermodynamics including different crystal growth or segregation/precipitate formation models potentially can condition DL models.

In this study, we demonstrate the applicability of deep learning (DL) for the segmentation of complex phase steel microstructures. Since its individual constituents differ only in shape

and arrangement of ferritic and carbon-rich phases rather than image intensity levels, traditional segmentation approaches reach their limits. We propose a holistic approach since the contrasting and imaging has pronounced implications for the DL methodology in terms of data imbalance, variance and spatial feature density. Amongst others, this includes annotations informed by electron backscatter diffraction to alleviate the burden of the manual annotation process based on how the microstructure in topography contrast micrographs visually appears to the expert eye. This allowed to provide a well-founded, objective annotation. While the segmentation models presumably benefit from more data, the trained U-Net networks achieved a satisfying performance from training with only 30–50 microscopic images. We hope that rebutting the general preconceptions about the large required data quantities, mitigates the reservations towards DL and ultimately encourages more scientists to research in this interdisciplinary field. The results point towards a general robustness of the U-Net with respect to modifications in the training procedure and architecture. Through the experimental design, a general guideline for the application of DL for microstructure inference could be derived. This applies in particular to the appropriate consideration of image context, data augmentation, imaging modalities, and pre-training. The network decisions to distinguish lath-bainite from its surroundings are visualized through the Grad-CAM and NetDissect methodologies. These suggest plausible and human comprehensible choices for features such as parallelism of inter-lath carbides, grain boundary junctions, grain aspect ratios and carbon-rich clusters. This is an important step towards the acceptance of DL segmentation in material science community. Finally, we provide an outlook on aspiring and auspicious cutting-edge methodologies from computer science that hold the potential to render microstructure inference from micrographs generalizable across materials and processes. A fundamental requirement to achieve this is the interoperability of diverse data generated across institutes. With the development of materials ontologies and the systematic digitalization of workflows, identifying and unifying relevant data across institutes will come within reach and thus increase the scope of such deep learning techniques substantially.

## Methods

**Material**. The material used in this study is a low-carbon CP steel, taken from industrially produced heavy plates. Steels were thermo-mechanically rolled, followed by controlled accelerated cooling. Figure 7 illustrates a plate cross section, where the lath-shaped bainite as well as polygonal and irregular ferrite with dispersed granular carbon-rich 2nd phase are highlighted.

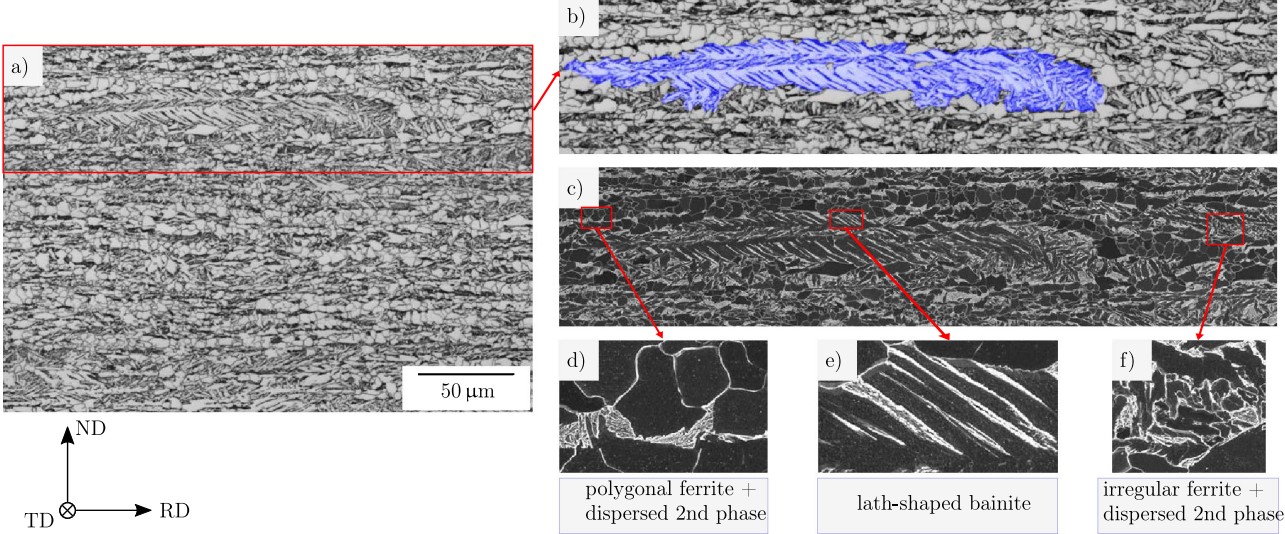

**Fig. 7 Overview of microstructure and contained phases. a** LOM micrograph of CP steel microstructure after Nital etching. **b** enlarged detail from (**a**), showing an annotated lath-shaped bainite region (blue). **c** correlative SEM micrograph of (**b**). The enlarged detail figures **d**–**f** depict polygonal ferrite with dispersed carbon-rich 2nd phase, lath-shaped bainite, and irregular ferrite with dispersed carbon-rich 2nd phase, respectively. In the SEM modality, the carbon-rich 2nd phase particles appear bright. RD, TD, and ND refer to the rolling, transverse and normal direction of the plate, respectively.

**Specimen preparation**. Specimens were taken in the plate's transverse direction (TD), ground using 80–1200 grid SiC papers, and then subjected to 6, 3, and finally, 1 μm diamond polishing. For LOM investigation, metallographic etching was carried out by immersing polished specimen surfaces into a mixture of ethanol and nitric acid (2 vol.-%), also called "Nital" etching. For SEM examination, the specimens were etched electrolytically using Struers electrolyte A2. Nital and electrolytic etching were chosen because they attack and thus reveal grain boundaries. This contrasting step is crucial to identify the boundaries of lath-shaped bainite regions during annotation. For investigation by electron backscatter diffraction, colloidal oxide polishing was additionally performed after diamond polishing.

**Microscopy**

*Light optical microscopy.* For imaging, the LOM in an Olympus LEXT OLS 4100 laser scanning microscope was used. Images were taken at a magnification of ×1000 with an image size of 1024 × 1024 pixels, corresponding to an area of 129.6 × 129.6 μm² (pixel size = 126.6 nm). All images were acquired with the same image contrast and brightness settings.

*Scanning electron microscopy.* SEM images were recorded in a Zeiss Merlin FEG-SEM using secondary electron contrast at a magnification of × 2000 with an image size of 2048 × 1536 pixels, equal to 56.7 × 42.5 μm² (pixel size = 27.7 nm). The SEM was operated at an acceleration voltage of 5 kV, a probe current of 300 pA, and a working distance of 5 mm. All images were acquired with the same image contrast and brightness settings in the SEM. During SEM image acquisition, lath-bainite regions were oversampled and are therefore overrepresented in the data.

*Correlative microscopy.* In a correlative approach, LOM and SEM were combined with EBSD characterization. The specimen regions of interest were marked by hardness indents for consistent imaging in different modalities. EBSD measurements were performed in a Zeiss Merlin FEG-SEM with an acceleration voltage of 25 kV, a probe current of 10 nA, and 15 mm working distance. Scans were done at a magnification of × 200 with a step size of 0.35 μm using a hexagonal grid. Data were analyzed using software OIM TSL Analysis. As cleanup, neighbor confidence index (CI) correlation (CI ≥ 0.01) and neighbor orientation correlation (5° grain tolerance angle) were applied. After EBSD measurements, specimens were etched, and micrographs from the same regions of interest were taken in LOM and SEM. Several such micrographs were stitched together using Microsoft Image Composite Editor to match the EBSD scanned region.

When combining different imaging techniques, the different micrographs must be aligned. This process is referred to as multi-modal image registration and is accompanied by challenges including different specimen states, viewpoints, contrasts, and fields of view[46]. For a general explanation of challenges during correlative characterization and image registration in metallography, the authors refer to Britz et. al.[47].

For registering EBSD maps with LOM and SEM images, the open-source tool ImageJ and its plugins SIFT feature extraction and bUnwarpJ registration were used[47]. First, the Scale-Invariant-Feature Transform (SIFT)[48] algorithm was used to find the same features in both the EBSD map and the LOM/SEM image. For this purpose, the EBSD image quality map[49] was chosen due to its pronounced similarity to the other

modalities. The common features extracted by SIFT facilitate the registration using the bUnwarpJ[50] algorithm. Thereby a transformation matrix is determined that is applied to register other EBSD-derived maps, e.g., misorientation maps.

**Data set preparation**

*Annotations for deep learning segmentation.* Labeling of images was done manually by human experts on a Wacom Tablet. Since human labeling based solely on the microstructure's visual appearance in topography-sensitive LOM or SEM images can be subjective, parameters from correlative EBSD measurements were used as additional information to annotate the micrographs more objectively. Reasonable EBSD-derived information that assisted the annotation included grain structure visualizations as well as intergranular and intragranular misorientation metrics. Namely, unique grain color, grain boundary, GOS, grain average misorientation (GAM), and KAM (with 3rd order neighbors) maps were considered.

Because of time constraints, it is typically not feasible to obtain high-fidelity annotations through correlative EBSD measurements for the comparatively large image sets required for DL. Therefore, correlative EBSD measurements were collected only for a subset of images, and the knowledge and experience gained from the fused data were translated to regular LOM and SEM images. For this reason, the correlative measurements can be regarded as references for the whole data set. Under these circumstances, well-founded and more objective annotations can be accomplished by human experts also without the EBSD data.

The final LOM image set consists of 51 micrographs with corresponding masks for the segmentation (1024 × 1024 pixels, ~28% lath-shaped bainite on average per image) and the final SEM image set of 36 micrographs with corresponding masks (2048 × 1433 pixels due to cropping of the SEM annotation bar, ~60% lath-shaped bainite on average per image).

*Data pre-processing for model training.* The raw input and derived label images were cropped into tiles for both imaging modalities to comply with network architectural and computational memory constraints. Two major variations were investigated with respect to data pre-processing—the influence of image downscaling (I) and tiling size (II).

I. Since the SEM images were acquired with higher magnification, the raw images covered a substantially smaller field of view. Tile sizes of 256 × 256 pixels and 512 × 512 pixels were selected for the LOM and SEM modality, respectively, to assimilate their contained image context to some degree. Before extracting tiles with the aforementioned fixed resolution, an optional downscaling step by a factor of × 0.5 in both spatial directions was performed to study the influence of image context, context passed into the receptive field, and information loss. So-called padding refers to the artificial extension of the image at its border. To ensure data-efficient tiling (3 × 2 tiles) and resolution conformity throughout the forward pass of the networks despite the native SEM resolution (2048 × 1433 pixels), mirror padding was applied at the top and bottom image border before extracting tiles. Owing to the dimensions of the native SEM image, its downscaled version was mirror padded comparatively more to facilitate 2 × 2 training and testing tiles of 512 × 512 pixels size. Tiles were cropped without overlap. For the downscaling study, four DL data sets were derived from the LOM and SEM raw data sets—native and downscaled versions of both image modalities.

**Table 3 Architectural differences between the vanilla U-Net and the U-Net VGG16.**

|  | Vanilla U-Net | U-Net VGG16 |
|---|---|---|
| Upsampling approach | Transposed convolutions (learnable) halving channel number | Bilinear Interpolation maintaining channel number |
| Convolution approach encoder | Contains two convolution layers in each level | Contains three convolution layers in later encoder and center levels |
| Batch normalization | Encoder and decoder | Decoder |

II. Intending to investigate the effect of the scale of phases with respect to the tile size, we created a set of differently tiled images from the native LOM images. Specifically, tiles of size $64 \times 64$, $96 \times 96$, $128 \times 128$, $192 \times 192$, $256 \times 256$, and $512 \times 512$ pixels were prepared. In this setting, as opposed to the downsizing experiments, information loss was absent, and physical pixel size is maintained.

In both studies, LOM input images were converted into grayscale. A summary of all tiled data sets, including some characteristic metrics, can be found in the Supplemental. Note that aside from the difference in raw image amount between the modalities, their different magnifications cause a large discrepancy in available data quantity in terms of physical specimen area. The resulting tiles were randomly sampled in five data portions for k-fold cross-validation (unstratified) with $k = 5$. For each of the five folds, the test set was altered to be one of the data portions. Hence, the five distinct data sets contained 80% and 20% of the total tile amount for training and testing, respectively.

While usually all models were evaluated on the testing sets with equivalent tile sizes and scale factors, the models for the tiling study were additionally evaluated on full resolution images to identify the impact of the provided field of view at the training stage.

**Deep learning methodology**. Two research groups collaborated on this segmentation task using their respective approaches and best practices. Both approaches are based on the U-Net architecture, but still contain some differences. By comparing their results, important conclusions regarding the universal applicability and robustness of deep learning techniques for the segmentation of CP steels can be drawn.

*Deep learning segmentation approach 1—Vanilla U-Net.* A vanilla U-Net model with an architecture implemented in the PyTorch framework[51] that included few adjustments with respect to ref. [19] was trained from scratch. Only two class channels in the output were used since the work at hand covers a binary segmentation problem. Furthermore, batch normalization was incorporated after convolutions to accelerate the training procedure by smoothing the optimization function[33]. The U-Net architecture has four levels, uses padding for the non-dilated $3 \times 3$ convolutions in the encoder, utilizes $2 \times 2$ max-pooling, applies "same" padding for transposed convolutions in the decoder, and contains skip-connections[52] between the corresponding encoder and decoder levels. A schematic of the architecture showing the designation of the individual layers is accessible in the Supplemental since network visualization techniques shown hereafter are referring to specific layers. Different online data augmentation techniques from the Albumentations package[53] were applied to investigate their impact on the performance. In contrast to our prior study[19], a more systematic approach to data augmentation was taken by applying grid, and random search for optimization of relevant hyperparameters in Tune[54]. The set of optimized augmentation parameters for both image modalities is outlined in the Supplemental. For training, the focal loss function[55], and an Adam optimizer[56] was used. Each model was trained for 250 epochs on a NVIDIA Tesla V100 GPU with 32GB memory and CUDA (v10.0) acceleration. The $\alpha$ and $\gamma$ parameters of the focal loss function were also considered during hyperparameter optimization to account for data set class imbalances.

*Deep learning segmentation approach 2—U-Net with VGG16 backbone.* A U-Net model variant with a VGG16 encoder that was pre-trained on ImageNet[57] was applied. During fine-tuning all layers were tuned simultaneously, the pre-trained encoder and the random initialized decoder. The model was implemented in PyTorch, and the pre-trained weights were from torchvision. A schematic of the architecture is depicted in the Supplemental. The initial five convolution blocks of VGG16 represent the four encoder levels and center level of the U-Net. The decoder contains four upsampling blocks. Each upsampling block contains a bilinear upsampling and two convolutional layers with batch normalization and Relu activation. Skip-connections are applied identically to the regular U-Net. In Table 3 relevant architectural differences are listed. Identical to the vanilla U-Net, the image shape is maintained by this model due to padded convolutions in the encoder. The major difference compared to segmentation approach #1 in the training procedure is that the U-Net VGG16 leveraged a pre-trained encoder. For performance optimization in the training process, online data augmentation (see Supplemental), cross-entropy loss, and an Adam optimizer[56] with cosine annealing schedule are utilized. Pre-trained models and random initialed validation models were confirmed to be converged after training for 150 and 250 epochs on a NVIDIA GeForce GTX 1080 Ti, respectively. Computational efficiency during training and testing depends on the hardware and implementation and is given as an indication in the

Supplemental despite omitting deployment optimization such as model pruning. With the exception of a validation experiment, no measures for the correction of class imbalance were taken.

The model performances are evaluated in terms of the *Accuracy* metric and the intersection over union (*IoU*), also referred to as Jaccard index, defined as follows:

$$Accuracy = \frac{TP + TN}{TP + TN + FP + FN} \tag{1}$$

$$IoU = \frac{TP}{TP + FP + FN} \tag{2}$$

Here, TP, TN, FP, and FN are the true positive, true negative, false positive, and false negative pixel amounts, respectively. Both metrics are defined in the range from zero to unity (or 0–100%), where latter corresponds to an ideal model prediction. The accuracy metric measures the correctly predicted pixel percentage, while the IoU measures the ratio between intersection and union of predicted and labeled pixel areas. We exemplary provide the accuracy metric due to its intuitiveness and despite its limited sensitivity in case of notable class imbalances, such as in the LOM case. In contrast, the IoU captures the model differences more adequately for data sets skewed towards the negative class, which is why we focus on it for the comparison between the individual models.

*Network visualization techniques.* In order to render model decisions explainable, the Network Dissection[58] and Gradient Weighted Class Activation Maps (Grad-CAM)[59] visualization methods were used. The objective of the Network Dissection method is to visualize concepts that were learned by individual filters in specific layers. This is achieved by evaluating activation maps, i.e., single channels of the activation function output, with regard to its spatial attention for regions in the input image. In particular, activation maps were thresholded such that the largest 1.0% of the activation map is obtained. In the original implementation, the thresholded activation maps were then resized to input image resolution and subsequently superimposed. However, since the encoder used unpadded convolutions, in the vanilla U-Net case, a combination of resizing and padding was required.

Grad-CAM, on the other hand, aims to shed light on the decision-making process of models. This technique originally focused on providing a class discriminative localization map for the output convolutional layer for a given input image, highlighting the important regions in the image for a particular class prediction. However, it is also applicable for any convolutional layer in a network. The localization map for a convolutional layer is constructed by a weighted combination of feature maps of that layer for a given input image. The weights for feature maps are computed by propagating the gradient of the particular class score with respect to the feature maps and performing a global average pooling over width and height dimensions. Since the method is applicable only for classification problems, we converted our network prediction to a classification output by global average pooling. Both methodologies for network visualization complement each other well and can, when combined, generate detailed insights into the decision-making process of DL architectures[3].

## Data availability
The datasets generated during and/or analyzed during the current study are not publicly available because they are part of an ongoing study and subject to third party (AG der Dillinger Hüttenwerke) restrictions. Source data are provided with this paper.

## Code availability
The codes used in this study are available after study completion (July 2022) from the authors upon reasonable request.

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

## Acknowledgements

The contribution of A.D. was funded by the Bosch-Forschungsstiftung im Stifterverband grant number T113/30074/17 and Open Access funding provided by project DEAL. The authors wish to acknowledge the EFRE Funds of the European Commission and the State Chancellery of Saarland for support of activities within the ZuMat project. The authors would also like to thank steel manufacturer "AG der Dillinger Hüttenwerke" for providing the sample material and acknowledge support by German Research Foundation (DFG, Deutsche Forschungsgemeinschaft). Work at Carnegie Mellon University was supported by the National Science Foundation under grant CMMI-1826218.

## Author contributions

Conceptualization: A.D., A.T., B.L., D.B., E.H. and M.M; Data curation: A.D. and M.M; Formal analysis: A.D., A.T. and B.L.; Investigation: A.D., A.T., B.L. and M.M.; Methodology: A.D., A.T., B.L. and M.M.; Project administration: A.D., B.L., D.B. and M.M.; Resources: C.E., D.B., F.M., E.H. and P.G.; Software: A.D., A.T. and B.L.; Supervision: C.E., D.B., F.M., L.H. and P.G.; Visualization: A.D., A.T., B.L. and M.M.; Writing—original draft: A.D., A.T., B.L., D.B. and M.M.; Writing—review & editing: all authors.

## Funding

## Competing interests

The authors declare no competing interests.

## Additional information

**Peer review information** *Nature Communications* thanks Gaetano Impoco and the other anonymous reviewer(s) for their contribution to the peer review this work. Peer reviewer reports are available.

