## [Peer Review File · Nature Communications]

Reviewers' Comments:

Reviewer #1:

Remarks to the Author:

Overall, I find the manuscript ready for submission. The authors have taken enough care in ensuring that the domain knowledge of materials scientists is successfully ported to use with emerging DL techniques. The sample preparation steps are extensively described and the model architecture and the training techniques are discussed well. Though I am not an expert in material science and cannot judge the novelty from that point of view, I do find the work to be a good introductory application of DL to the material sciences domain. I have a few questions for the authors as listed below:

Pg. 9: "In our study, we successfully trained both random initialized networks and pre-trained networks with comparatively small data sets of approximately 50 and 30 images for LOM and SEM, respectively. This invalidates the general claim of DL being only applicable for large-scale data sets" -- Note that the validation sets in both cases are extremely small compared to DL standards. The authors validate the current work with LOO cross validation which does not reveal the true ability of the model to generalize, in that the model could be overfitting to the limited evidence. There seem to be not much distribution shift between the training and test sets.

Vanilla U-net scoring better than U-Net VGG16:

1. It is possible that this is the result of the domain of imagenet being very different than the materials. Have the authors tried finetuning more layers of the pretrained network instead of only the final layer? There is usually a tradeoff in terms of the number of layers finetuned vs the resulting performance.
2. The authors must apply the focal loss class balancing to finetune the pretrained model. Keep the loss fixed and change the models alone to verify the difference.
3. The difference between LOM/SEM cases is puzzling given that the difference in # of data points isn't by a large factor.

Reviewer #2:

Remarks to the Author:

The paper addresses the problem of highlighting phases in (optical and SEM) micrographs of compound materials. The uneven distribution of grey-levels into each phase defies segmentation algorithms that rely solely on pixel intensity. The authors adopt two slightly different deep learning approaches and compare them against manually-annotated golden standard. The main objective is to show that U-Net architectures are a reliable tool to solve such complex problems with no need of fine tuning. The authors claim that this is demonstrated by the very fact that different flavors of these architectures yield similar results and that performance variations mostly depend on data manipulation (e.g., tiling and padding) rather than on architectural differences. Indeed, the results shown by the authors are quite good for the problem at hand, even more so due to the reduced size of the training set (quite common in material sciences).

This paper does not describe a novel approach to image segmentation. However, the multidisciplinary approach of this work to the segmentation problem, encompassing sample preparation, image acquisition and image segmentation, is certainly significant (and, I would say, uncommon). It points towards a direction that all future research in this field should follow. The paper thoroughly addresses several open questions about segmentation and deep-learning, including the interpretation of the inner activity of the network, that is, one of the most controversial aspects of deep learning approaches. It is also interesting in one more respect: two research groups work on the same problem to demonstrate the validity of a framework, rather than the quality of a single approach. In this sense, I think that this paper is a useful contribution to the scientific debate about the role of deep learning in industrial applications.

The paper clearly presents speculative thinking about each and every step of network training and operativity. The discussion is supported by experimental evidence and a sufficiently rigorous analysis of data. The description is reasonably detailed. A few details could be added, e.g., about the operators used in the network to process images as well as about the size of padding and overlap between image tiles. It would be interesting to compare the size of padding to a

representative measure of the size of phases.

The conclusions are consistent with experimental data. I would suggest to think a little bit more about a few statements, such as the claim that the small size of the training dataset used in experiments, compared to the quality of results, "invalidates the general claim of DL being only applicable for large-scale data sets". This is certainly true for the dataset used in this paper. However, this can well prove false for other data sets. The successive claim that the availability of reproducible, high-quality imaging would reduce the need for a large amount of training data appears a leap of logic.

At the end of page 9, the authors write that the models fail in the very same regions where human expert make mistakes during manual annotation. No further explanation is given in order to better understand the method used to locate these regions. Do the authors refer, e.g., to pixel regions for which manual annotations do not match?

Thinking about the possibility to generalize the results shown in the paper, I would appreciate a deeper discussion about how much the performance of DL methods is affected by the scale of phases in the image. Similarly, in order to accept the claim that these methods are generalizable, I would like to see a few comparative experiments with radically different datasets.

Finally, the authors do not address execution time. This is an important aspect for industrial applications, especially in manufacturing.

Reviewer #3:

Remarks to the Author:

This paper purports to lay out a holistic approach to deploying deep learning for complex microstructure inference. This is a very topical and challenging problem, and I really wanted to like this paper. However the paper as it stands does not accomplish any of the key claims. I lay out my reasons below:

- 1) Claim of impact: Showing segmentation on one class of microstructures is not enough (according to this reviewer). This is insufficient to illustrate their central claim of "an intuition about the required data quality and quantity and an extensive methodological DL guideline for microstructure quantification and classification are still missing". No intuition is gained given the limited number of models used, and the limited data used (see point #3).
- 2) There are several other real-world images (for instance, MRI images) that are as (if not more) complex than microstructure images, and a lot more critical to segment correctly. This goes right to the heart of the statement "Furthermore, microstructure recognition tasks, compared to real-world images, can be very complex regarding ..."
- 3) Data size: The datasize used (30-40) images makes the results of this study highly suspect. It was not clear to me how many images were finally used after augmentation and windowing, but training a U-Net from scratch is not a good idea with this few images. See for instance <https://arxiv.org/pdf/2001.05566.pdf> .
- 4) Transfer learning: As the author's comment in outlook section, using pre-training with self- or semi-supervised learning is the way to go with such small datasets. The marginal improvements of using a pre-trained U-Net is indeed indicative of this. See for instance "Zhuang F, Qi Z, Duan K, Xi D, Zhu Y, Zhu H, Xiong H, He Q. A comprehensive survey on transfer learning. Proceedings of the IEEE. 2020 Jul 7;109(1):43-76."
- 5) Transparent decision making: Grad-CAM has been shown to be a very poor explainability mechanism. See for instance <https://arxiv.org/abs/1812.02843> where GradCAM giver poor explainability even when the model predictions are accurate.
- 6) Intra- and Inter-rater variability: This was an opportunity lost to discuss and provide intuition on inter- and intra- rater variability during the human annotation. That is, do changes in the fg and bg as marked by experts significantly impact results?
- 7) Metrics: Using accuracy is a bad metric. In some cases, the pixel count of fp to bg is 1:3 or 1:4

which always segmenting as bg will give 75-80% accuracy. A 10% improvement seems marginal. I encourage sticking to IoU as the key figure of merit.

8) Lack of detailed analysis: While the authors do a good job of performing ablation studies, this was an opportunity lost with showing extensive analysis that could back up (at least anecdotally) some of their claims. Specifically, what happens when depth of the U-net is changed, what happens when window size (and hence data size) changes, what happens when total data set is further reduced, how is pixel size, window size and resolution related to performance. All these would add value and generate more intuition for practitioners.

9) Data sharing: Without the availability of annotated data for the broader audience to try and evaluate, replicate and improve on these (standard) methods, I see the impact of this paper as minimal.

Reviewer #1 (Remarks to the Author):

Overall, I find the manuscript ready for submission. The authors have taken enough care in ensuring that the domain knowledge of materials scientists is successfully ported to use with emerging DL techniques. The sample preparation steps are extensively described and the model architecture and the training techniques are discussed well. Though I am not an expert in material science and cannot judge the novelty from that point of view, I do find the work to be a good introductory application of DL to the material sciences domain. I have a few questions for the authors as listed below:

Pg. 9: "In our study, we successfully trained both random initialized networks and pre-trained networks with comparatively small data sets of approximately 50 and 30 images for LOM and SEM, respectively. This invalidates the general claim of DL being only applicable for large-scale data sets" -- Note that the validation sets in both cases are extremely small compared to DL standards. The authors validate the current work with LOO cross validation which does not reveal the true ability of the model to generalize, in that the model could be overfitting to the limited evidence. There seem to be not much distribution shift between the training and test sets.

Author response:

Thank you for your efforts in reviewing our work.

You are right that the validation sets are small. The datasets were acquired in a very reproducible fashion and do not contain the typical variances that are otherwise introduced in metallography through multiple operators, multiple microscopes or different etchings. A strong indication is that the applied data augmentation did not improve the performance substantially despite the small size of the dataset.

In our section on „Variances and generalization“ on page 11 we address this:

„In instances where such material-extrinsic variance can be ensured to be insignificant, data augmentation through simple spatial (affine and even elastic) or intensity transformations can be evaded. Therefore, such models trained on comparatively small data sets are suitable for tasks with inherently small scatter, such as quality inspection, where recurring tasks and predefined workflows are set. When, for instance, etching-based contrasting methodologies are concerned, reproducibility can be difficult to attain. “

Subsequently we show that the model fails when we apply it (which achieved good accuracy in the source domain) on an alternate set of micrographs where material was etched with different parameters. As we highlight in the manuscript, this can be ascribed to the low-quantity data with small material processing or imaging-induced variance. We show that if we use brightness and contrast augmentation, the generalization with respect to the otherwise etched domain improves. We want to stress that the dataset was not optimized for achieving generalization over a wide range of processing routes, materials or similar but to perform an ablation study helping material scientists to get an idea about how different approaches (hyperparameter and rescaling and so on) affects the training.

Vanilla U-net scoring better than U-Net VGG16:

1. It is possible that this is the result of the domain of imagenet being very different than the materials. Have the authors tried finetuning more layers of the pretrained network instead of only the final layer? There is usually a tradeoff in terms of the number of layers finetuned vs the resulting performance.
2. The authors must apply the focal loss class balancing to finetune the pretrained model. Keep the loss fixed and change the models alone to verify the difference.
3. The difference between LOM/SEM cases is puzzling given that the difference in # of data points isn't by a large factor.

Author response:

Add 1:

That the U-Net scoring is better than the pretrained U-Net VGG16 in the optical light microscopy case was in fact an artifact. Since we used unpadded convolutions in the vanilla U-Net and the tiles were extracted with an overlap but train and test tiles were sampled from the same raw images we had a mixing of training and testing data. This error did not apply to the U-Net VGG16 as it used padded convolutions and center-cropping (removing the overlap region) before passing the data to the network. We resolved this issue by applying the same procedure (center-cropping and padded convolutions) in the vanilla U-Net and conducting the experiments again. As a positive side effect, this increased the comparability between the two approaches.

Add 2:

Thank you for your comment on fine-tuning. We did finetune over the full network in this case since the initial submission. We did add a remark on how we finetuned in the methods section.

Despite identifying this as an artifact, we did implement the identical loss function for the VGG16 network and the fine tuning training according to your remark. However, assimilating the loss functions did not affect the results significantly. We added this as a statement in the results and added more notes on the loss functions in the methods section applied.

Regarding point 3: In fact, the data amount is substantially different (not only the image amount 51 vs 36) but especially the physical image size resolution and the characterized area in μm^2 differs significantly. As the features have the same size independent of the microscopy methodology, the LOM contains a much larger number of grains. Therefore, we argue that the tendencies are well explainable. In the light optical microscopy micrographs, the effect of pre-training is smaller since more data is available, while in the SEM models the pretraining has a major impact.

Reviewer #2 (Remarks to the Author):

The paper addresses the problem of highlighting phases in (optical and SEM) micrographs of compound materials. The uneven distribution of grey-levels into each phase defies segmentation algorithms that rely solely on pixel intensity. The authors adopt two slightly different deep learning approaches and compare them against manually-annotated golden standard. The main objective is to show that U-Net architectures are a reliable tool to solve such complex problems with no need of fine tuning. The authors claim that this is demonstrated by the very fact that different flavors of these architectures yield similar results and that performance variations mostly depend on data manipulation (e.g., tiling and padding) rather than on architectural differences. Indeed, the results shown by the authors are quite good for the problem at hand, even more so due to the reduced size of the training set (quite common in material sciences).

This paper does not describe a novel approach to image segmentation. However, the multidisciplinary approach of this work to the segmentation problem, encompassing sample preparation, image acquisition and image segmentation, is certainly significant (and, I would say, uncommon). It points towards a direction that all future research in this field should follow. The paper thoroughly addresses several open questions about segmentation and deep-learning, including the interpretation of the inner activity of the network, that is, one of the most controversial aspects of deep learning approaches. It is also interesting in one more respect: two research groups work on the same problem to demonstrate the validity of a framework, rather than the quality of a single approach. In this sense, I think that this paper is a useful contribution to the scientific debate about the role of deep learning in industrial applications.

The paper clearly presents speculative thinking about each and every step of network training and operativity. The discussion is supported by experimental evidence and a sufficiently rigorous analysis of data. The description is reasonably detailed.

A few details could be added, e.g., about the operators used in the network to process images as well as about the size of padding and overlap between image tiles. It would be interesting to compare the size of padding to a representative measure of the size of phases.

The conclusions are consistent with experimental data. I would suggest to think a little bit more about a few statements, such as the claim that the small size of the training dataset used in experiments, compared to the quality of results, “invalidates the general claim of DL being only applicable for large-scale data sets”. This is certainly true for the dataset used in this paper. However, this can well prove false for other data sets. The successive claim that the availability of reproducible, high-quality imaging would reduce the need for a large amount of training data appears a leap of logic.

Author response:

Thank you for your detailed and positive review. The interdisciplinary collaboration is a fundamental necessity to advance the implementation of DL into the materials science area. Furthermore, the industrial usage requires convincing real world examples and usable frameworks. As you also pointed out, the real world of materials science and engineering works on a lot of small data cases and therefore we have targeted this as a priority.

Nevertheless, we would like to point out that fine tuning of pre-trained networks improves performance. In fact, we believe that in this low data regime, that we are in right now, pre-training and fine-tuning are of substantial relevance. In both LOM and especially SEM we show that an improvement through pretraining can be achieved. We enhanced our manuscript in the discussion to assure not to be suggestive of pre-training being unuseful.

“invalidates the general claim of DL being only applicable for large-scale data sets” → We used „general“ here to emphasize that it is a preconception that many (material) scientists have, who are not involved with data-driven techniques. We believe that in the setting of small variance data (reproducible image acquisition) fewer data is necessary to train a model for that domain. However, we observed that such a model won't be great at generalizing if trained with small processing-induced variance data. We understand this and hence the successive remark has been adjusted in the discussion.

At the end of page 9, the authors write that the models fail in the very same regions where human expert make mistakes during manual annotation. No further explanation is given in order to better understand the method used to locate these regions. Do the authors refer, e.g., to pixel regions for which manual annotations do not match?

Author response:

You are right, we added few sentences here to render in more clear in the discussion.

Thinking about the possibility to generalize the results shown in the paper, I would appreciate a deeper discussion about how much the performance of DL methods is affected by the scale of phases in the image. Similarly, in order to accept the claim that these methods are generalizable, I would like to see a few comparative experiments with radically different datasets.

Author response:

We added a study on relative scale of phases and image size to the manuscript. Specifically we conducted another study where we altered the size of the tiles in LOM (rather than downscaling so we could exclude information loss and receptive field based effects). This study gives more insights on how to choose the tile sizes ideally, as it correlates tile sizes with lath width and lath-bainite region size distributions. Accordingly, we extended the results (Image context and network receptive field dependency section) and discussion (Image context and network receptive field dependency section).

On the other hand, we did not claim that the model trained with this low-variance dataset is generalizable across data sets. In fact, we did apply the trained models to an otherwise etched (overetched) image and observed that the model is not so good at generalizing across domains (see section „Variances and generalization“). Even when adjusting and optimizing the data augmentation to lower the domain gap towards a target domain, the model does not generalize particularly well. We stress that, our objective in this paper was not to train a perfect model for cross-domain generalization but rather to highlight image processing effects (scaling, tiling...) and pretraining on DL performance as well as explainability. If generalization would have been our primary objective, we would have acquired our dataset with a more natural variance and would have applied stronger data augmentation.

Finally, the authors do not address execution time. This is an important aspect for industrial applications, especially in manufacturing.

Author response:

We added notes on execution time both for training and for deployment in the Supplemental. We added number of trained epochs and specified hardware as well in the methods section (Deep learning segmentation approach section).

Another important note to the reviewer:

The U-Net scoring better than the pretrained U-Net VGG16 in the light microscopy case was an artifact. Since we used unpadded convolutions in the vanilla U-Net and the tiles were extracted with an overlap but train and test tiles were sampled from the same raw images we had a mixing of training and testing data. This error did not apply to the U-Net VGG16 as it used padded convolutions and center-cropping (removing the overlap region) before passing the data to the network. We resolved this issue by applying the same procedure (center-cropping and padded convolutions) in the vanilla U-Net and conducting the experiments again. As a positive side effect, this increased the comparability between the two approaches.

Reviewer #3 (Remarks to the Author):

This paper purports to lay out a holistic approach to deploying deep learning for complex microstructure inference. This is a very topical and challenging problem, and I really wanted to like this paper. However the paper as it stands does not accomplish any of the key claims. I lay out my reasons below:

1) Claim of impact: Showing segmentation on one class of microstructures is not enough (according to this reviewer). This is insufficient to illustrate their central claim of “an intuition about the required data quality and quantity and an extensive methodological DL guideline for microstructure quantification and classification are still missing”. No intuition is gained given the limited number of models used, and the limited data used (see point #3).

Author response:

Thank you for your efforts in reviewing our work.

You are right, the claim is too strong. Nevertheless, this manuscript clearly indicates that data quantities of 30-50 images suffice to train a segmentation network for this low material-extrinsic variance domain. Please note that both data sets have quite different data amounts, since, aside from the image number, individual LOM images capture a substantially larger physical area. In addition, when it comes to image rescaling and pretraining, we firmly believe that our experiments after the corrections provide a valid indication. Namely, that increasing the context and receptive field helps when dealing with such long-range features and by how much pretraining is helping in both distinct data quantity settings (low-quantity datasets are very typical for materials science data sets). It is true that we do not perform a study on data quality in this manuscript. We removed this claim and softened the claim concerning data quantity. We adjusted the abstract and introduction accordingly.

2) There are several other real-world images (for instance, MRI images) that are as (if not more) complex than microstructure images, and a lot more critical to segment correctly. This goes right to the heart of the statement “Furthermore, microstructure recognition tasks, compared to real-world images, can be very complex regarding ...”

Author response:

By real-world images we are rather referring to street scenes (Cityscapes) or Image-Net, and similar. We slightly modified the wording from „real-world images“ to „natural images“ and added Image-Net as an example in the introduction to cause less confusion.

3) Data size: The datasize used (30-40) images makes the results of this study highly suspect. It was not clear to me how many images were finally used after augmentation and windowing, but training a U-Net from scratch is not a good idea with this few images. See for instance <https://arxiv.org/pdf/2001.05566.pdf> .

Author response:

We refer to the number of individual training tile images in the Supplemental. On page 4, we state „A summary of the data sets, including some characteristic metrics, can be found in the Supplemental.“ We adjusted the wording here to make it more clear. The amount of training tiles varies depending on the preprocessing and the exact data set. We kept it in the Supplemental rather than the manuscript since it would add large tables to an already somewhat long manuscript and since we consider the individual tile amount as not as important as the overarching raw microscope image number from which the tiles originate. For your convenience: In the non-resized LOM training, we worked with approximately 600 tiles.

In terms of augmentation we did apply online augmentation which means the number of epochs (which we now added consistently in the Methods Deep Learning section) and the dataset sizes (given in the Supplemental) directly provide the number of images which the network has seen.

We are aware that pretraining is a common and established practice in deep learning. However, we still wanted to make the comparison between ImageNet pretrained networks and random initialized ones. Finding

appropriate pretraining large-scale data sets with a small domain gap to the target task is difficult in materials science, which is why many practitioners do not apply pretraining currently.

4) Transfer learning: As the author's comment in outlook section, using pre-training with self- or semi-supervised learning is the way to go with such small datasets. The marginal improvements of using a pre-trained U-Net is indeed indicative of this. See for instance "Zhuang F, Qi Z, Duan K, Xi D, Zhu Y, Zhu H, Xiong H, He Q. A comprehensive survey on transfer learning. Proceedings of the IEEE. 2020 Jul 7;109(1):43-76."

Author response:

Thank you for sharing this information. In follow-up publications we aim to include semi-supervised learning and unsupervised domain adaptation more.

5) Transparent decision making: Grad-CAM has been shown to be a very poor explainability mechanism. See for instance <https://arxiv.org/abs/1812.02843> where GradCAM gives poor explainability even when the model predictions are accurate.

Author response:

It might be true that GradCAM, as other visualization techniques, is prone to such adversarial attacks. However, GradCAM was shown to provide comparatively sensible visualizations here (→) when exposed to sanity checks.

Adebayo, Julius, et al. "Sanity checks for saliency maps." *arXiv preprint arXiv:1810.03292* (2018). Sanity Checks for Saliency Maps (nips.cc)

Similarly, for our results the visualizations agree with expert annotators' expectations. From our point of view, adversarial attacks are not relevant in our setting.

6) Intra- and Inter-rater variability: This was an opportunity lost to discuss and provide intuition on inter- and intra-rater variability during the human annotation. That is, do changes in the fg and bg as marked by experts significantly impact results?

Author response:

We expect there to be substantially less rater-variability in annotation as annotation was assisted by correlative EBSD (orientation-sensitive data). We did an experiment in the beginning of our collaboration where we considered two data sets:

- data set where we discarded tiles with uncertain regions
- data set with all tiles

We did not see a major difference in performance. This we added as a remark in the manuscript in the result section.

7) Metrics: Using accuracy is a bad metric. In some cases, the pixel count of fp to bg is 1:3 or 1:4 which always segmenting as bg will give 75-80% accuracy. A 10% improvement seems marginal. I encourage sticking to IoU as the key figure of merit.

Author response:

Initially we wanted to provide accuracy as an additional metric because it is more intuitive than IoU. However, as you mention we have class imbalance in the light microscopy data set as it is representative for the microstructure. Therefore, omitting true negatives in IoU renders this metric more sensitive. Even though we explained these aspects in the paper for our materials science readers, we removed the accuracy values from the tables and just added a single statement in the result section that the IoUs correspond to approximately X accuracy.

Side note: We have in average 73% percent background in the light microscopy, so it is almost 20% increase over the naïve baseline.

8) Lack of detailed analysis: While the authors do a good job of performing ablation studies, this was an

opportunity lost with showing extensive analysis that could back up (at least anecdotally) some of their claims. Specifically, what happens when depth of the U-net is changed, what happens when window size (and hence data size) changes, what happens when total data set is further reduced, how is pixel size, window size and resolution related to performance. All these would add value and generate more intuition for practitioners.

Author response:

We added a study on relative scale of phases and image. To be specific, we conducted another study where we altered the size of the cropped LOM tiles (rather than downscaling, so we could exclude information loss and receptive field based effects). This addresses your window size remark. This study gives more insights on how to choose the tile sizes ideally as we compare the tile sizes with characteristic length scales (lath width or region size). Accordingly, we extended the results and discussion.

9) Data sharing: Without the availability of annotated data for the broader audience to try and evaluate, replicate and improve on these (standard) methods, I see the impact of this paper as minimal.

Author response:

Although we would like to share the data to fuel a similar multiplication effect as observed in other domains, in this case we are limited due to industry involvement. The data is part of an ongoing study.

Another important note to the reviewer:

The U-Net scoring better than the pretrained U-Net VGG16 in the light microscopy case was an artifact. Since we used unpadded convolutions in the vanilla U-Net and the tiles were extracted with an overlap but train and test tiles were sampled from the same raw images we had a mixing of training and testing data. This error did not apply to the U-Net VGG16 as it used padded convolutions and center-cropping (removing the overlap region) before passing the data to the network. We resolved this issue by applying the same procedure (center-cropping and padded convolutions) in the vanilla U-Net and conducting the experiments again. As a positive side effect, this increased the comparability between the two approaches.

Reviewers' Comments:

Reviewer #1:

None

Reviewer #2:

Remarks to the Author:

The authors addressed all the issues and concerns in my review. For that part (i.e., the point of view of a user), I think that their article has been improved with respect to the initial manuscript. If other reviewers share this thought, I think that the manuscript is ready for publication.

Just one more comment on fine-tuning that is not intended to influence the discourse in final paper. Perhaps, the authors will find it a useful thought for further investigation.

I believe that one should be very careful with tuning, at least for two reasons:

1. it often requires care, time, and competence that industrial applications cannot afford
2. there is always a risk of overtraining, especially with small datasets

Reviewer #3:

Remarks to the Author:

I thank the authors for the additional effort in responding to (some of) my questions. The manuscript has improved with the additional work, as well as the thoughtful watering down of some of the claims.

However, I do note that the watering down of several claims does bring down the overall impact of the paper. Several of my concerns (#2,#3,#7,#9) remain unanswered.

I remain unsatisfied with the key claims, especially when there are similar reports from other fields (for instance, the medical field) where complex images are successfully segmented. I also remain concerned with the small dataset used here. Finally, while understandable, the unavailability of the data makes comparative assessment impossible, further diminishing the impact.

Reviewer #2 (Remarks to the Author):

The authors addressed all the issues and concerns in my review. For that part (i.e., the point of view of a user), I think that their article has been improved with respect to the initial manuscript. If other reviewers share this thought, I think that the manuscript is ready for publication.

Just one more comment on fine-tuning that is not intended to influence the discourse in final paper. Perhaps, the authors will find it a useful thought for further investigation.

I believe that one should be very careful with tuning, at least for two reasons:

1. it often requires care, time, and competence that industrial applications cannot afford
2. there is always a risk of overtraining, especially with small datasets

Thank you for your review and remarks. We agree with your comments. In our case, fine-tuning was applied in a careful way (with optimized learning rates) and overfitting was not observed in the learning curves.

Reviewer #3 (Remarks to the Author):

I thank the authors for the additional effort in responding to (some of) my questions. The manuscript has improved with the additional work, as well as the thoughtful watering down of some of the claims.

However, I do note that the watering down of several claims does bring down the overall impact of the paper. Several of my concerns (#2,#3,#7,#9) remain unanswered.

I remain unsatisfied with the key claims, especially when there are similar reports from other fields (for instance, the medical field) where complex images are successfully segmented. I also remain concerned with the small dataset used here. Finally, while understandable, the unavailability of the data makes comparative assessment impossible, further diminishing the impact.

Thank you for your review and remarks. We agree that some works in other domains (e.g. medical) share similar observations with our work. However, we cited some of those in our manuscript, e.g.

Sabottke, C. F. & Spieler, B. M. The Effect of Image Resolution on Deep Learning in Radiography. Radiol. Artif. Intell. 2, e190015, DOI: 10.1148/ryai.2019190015 (2020).

We believe that the observed scatter in the data is predominantly materials' microstructure-based, since image acquisition was performed in a very reproducible fashion. Under this assumption and considering the characteristic sizes of microstructural features (such as lath width and grain sizes), it is very likely that the imaged area is representative of the microstructural scatter. We added this in a few sentences to make clear, why we achieve good results despite the low data quantity.